# Unique DUOX2+ACE2+ small cholangiocytes are pathogenic targets for primary biliary cholangitis

Xi Li[1,2,14], Yan Li[1,14], Jintao Xiao[1,3,14], Huiwen Wang[1,4,14], Yan Guo[1,5,14], Xiuru Mao[1,4,14], Pan Shi[1,14], Yanliang Hou[1,3], Xiaoxun Zhang[1], Nan Zhao[1], Minghua Zheng [6], Yonghong He[1], Jingjing Ding[1], Ya Tan[1], Min Liao[1], Ling Li[1], Ying Peng[1], Xuan Li[1], Qiong Pan [1], Qiaoling Xie[1], Qiao Li[1], Jianwei Li[7], Ying Li[8], Zhe Chen[9], Yongxiu Huang[9], David N. Assis[10], Shi-Ying Cai[10], James L. Boyer[10], Xuequan Huang [11] ✉, Can-E Tang [12,13] ✉, Xiaowei Liu [3] ✉, Shifang Peng [4] ✉ & Jin Chai [1] ✉

Cholangiocytes play a crucial role in bile formation. Cholangiocyte injury causes cholestasis, including primary biliary cholangitis (PBC). However, the etiology of PBC remains unclear despite being characterized as an auto-immune disease. Using single-cell RNA sequencing (scRNA-seq), fluorescence-activated-cell-sorting, multiplex immunofluorescence (IF) and RNAscope analyses, we identified unique DUOX2+ACE2+ small cholangiocytes in human and mouse livers. Their selective decrease in PBC patients was associated with the severity of disease. Moreover, proteomics, scRNA-seq, and qPCR analyses indicated that polymeric immunoglobulin receptor (pIgR) was highly expressed in DUOX2+ACE2+ cholangiocytes. Serum anti-pIgR autoantibody levels were significantly increased in PBC patients, regardless of positive and negative AMA-M2. Spatial transcriptomics and multiplex IF revealed that CD27+ memory B and plasma cells accumulated in the hepatic portal tracts of PBC patients. Collectively, DUOX2+ACE2+ small cholangiocytes are pathogenic targets in PBC, and preservation of DUOX2+ACE2+ cholangiocytes and targeting anti-pIgR autoantibodies may be valuable strategies for therapeutic interventions in PBC.

Primary biliary cholangitis (formerly called primary biliary cirrhosis, PBC) is a chronic autoimmune liver disease that is predominantly seen in women. Currently, there are over 100,000 new cases diagnosed annually in the world. Pathologically, PBC initially destroys biliary epithelial cells (BECs) in small intrahepatic bile ducts and eventually damages other types of liver cells, leading to progressive cholestasis, and resulting in liver cirrhosis and liver failure when untreated[1]. Clinically, PBC patients usually develop anti-mitochondrial antibodies (AMAs) though a minority are AMA negative[2]. Previous studies have

suggested that genetic and environmental factors, dysregulated immunity, and other factors contribute to the pathogenesis of PBC[1–3]. Loss of B cell tolerance to the E2 subunit of pyruvate dehydrogenase complex (PDC-E2) in the mitochondria is crucial for the development of AMA-M2 autoantibodies in PBC patients[1–3]. Furthermore, antigen-specific CD4+ and CD8+ T cells can migrate and accumulate in the portal tracts of PBC patients[4–6]. Hence, progressive humoral and cel-lular immune responses can lead to chronic inflammation and destroy BECs[1–6]. Given that PDC-E2 is widely expressed in liver cells and BECs

A full list of affiliations appears at the end of the paper. ✉e-mail: hxuequan@163.com; tangcane@csu.edu.cn; liuxw@csu.edu.cn; sfp1988@csu.edu.cn; jin.chai@cldcsw.org

are heterogeneous in phenotype[1–3], it remains unclear why autoimmune responses preferably destroy small BECs in the liver of PBC patients regardless of the presence of AMAs, and which BECs are targeted in the early stages of PBC pathogenesis.

Although PBC has autoimmune features, intrahepatic cholestasis due to cholangiocyte injury is characteristic of PBC[7,8]. Indeed, PBC patients do not respond to immunosuppressive agents alone[2]. Instead, most patients respond to ursodeoxycholic acid (UDCA), a first-line drug for cholestasis[7,8]. Polymeric immunoglobulin receptor (pIgR) is widely expressed in the basolateral membrane of epithelial cells, and is responsible for transport of immunoglobulin (Ig) A and IgM into the mucosal lumen[9,10]. Notably, hepatic pIgR is exclusively expressed in cholangiocytes[9]. Interestingly, pIgR-mediated transcytosis of PDC-E2 specific IgA can cause BEC apoptosis and biliary injury[10]. However, there are no reports on whether or how pIgR contributes to the development and progression of PBC, particularly in the early stages of PBC pathogenesis.

In this study, we have identified a unique type of DUOX2+ACE2+ small cholangiocytes in human and mouse livers (18.7% of human cholangiocytes and 22.2% of mouse cholangiocytes), using single-cell RNA sequencing (both 5′- and 3′-scRNA-seq), fluorescence-activated cell sorting (FACS), multiplex immunofluorescence (IF) and RNAscope methods. We have also studied their functional and clinical significance in PBC patients by spatial transcriptomics (ST) and proteomics. Our data demonstrate that DUOX2+ACE2+ small cholangiocytes that highly express pIgR antigen are pathogenic targets of PBC. Our results may provide insights into the early pathogenic process of PBC and uncover specific biomarkers for the diagnosis and therapeutic interventions for PBC.

## Results

### Single-cell transcriptomic atlas of human livers with and without PBC

We performed both 5′- and 3′-scRNA-seq analyses of liver cells from four control subjects and five PBC patients before UDCA treatment (Fig. 1a, Supplementary Fig. 1 and Supplementary Tables 1 and 2). The 5′-scRNA-seq data indicated that 70,050 liver cells were obtained from 4 control subjects (29,778 cells) and 5 PBC patients (40,272 cells) after quality control (Supplementary Fig. 2 and Supplementary Table 3). After dimension reduction and clustering analysis, uniform manifold approximation and projection (UMAP) revealed 30 cell populations (Fig. 1b and Supplementary Data 1), each containing cells from control and PBC livers (Fig. 1c) across 11 cell lineages (Fig. 1d, Supplementary Fig. 2a–c and Supplementary Data 2). Those cell lineages expressed their specific marker genes (Fig. 1e, f and Supplementary Table 4). The distribution of cell lineages within groups was highly reproducible (Supplementary Fig. 2d–f). Interestingly, remarkable differences in subpopulation distribution between the control and PBC livers were found (Fig. 1g, Supplementary Fig. 2d and Supplementary Table 5). As shown in Fig. 1h, the fractions of endothelial cells, cholangiocytes and mesenchymal cells decreased, while the proportions of T and plasma cells increased in PBC livers, compared to the control livers. Similar data were achieved by 3′-scRNA-seq (Supplementary Figs. 3 and 4). In addition, we also found some differentially expressed genes (DEGs) of each subpopulation of liver cells between the control and PBC groups (Supplementary Fig. 5).

### A unique type of DUOX2+ACE2+ small cholangiocytes are identified in human and mouse livers

Because PBC is characterized by selective destruction of small BECs[1–3], we paid close attention to cholangiocytes. Clustering analysis of 5′-scRNA-seq data revealed that cholangiocytes formed 8 clusters (1) to (8) in all subjects (Fig. 2a, b and Supplementary Data 3). Interestingly, a unique cholangiocyte cluster (3) accounting for 18.6% of all cholangiocytes was identified in human

control livers whereas this cluster of cholangiocytes was undetectable in PBC livers (Fig. 2c, d and Supplementary Tables 6 and 7). Remarkably, high levels of dual oxidase 2 (DUOX2) and angiotensin-converting enzyme 2 (ACE2) expression were detected in cluster (3) of cholangiocytes, but they were very low in other clusters of cholangiocytes (Fig. 2e, f), indicating a distinct subpopulation of cholangiocytes in human liver and a potential target for PBC. Similar data were obtained by analysis of the 3′-scRNA-seq data of human livers (Supplementary Fig. 6). Moreover, the numbers of cholangiocytes in clusters (1) and (2) in PBC patients decreased, compared to controls (Supplementary Table 6). However, there was no significant difference in the proportion of these clusters of cholangiocytes in total numbers of cholangiocytes in this population (Fig. 2d). In addition, other types of cells in the liver, such as immune cells (Supplementary Fig. 7 and Supplementary Data 4–6), endothelial and mesenchymal cells (Supplementary Fig. 8 and Supplementary Data 7, 8) also displayed varying heterogeneity.

Next, FACS analyses using anti-CK19, anti-DUOX2 and anti-ACE2 antibodies and their corresponding fluorescent isotype control antibodies indicated that primary CK19+DUOX2+ACE2+ cells accounted for 18.7% of CK19+ liver cells in human control livers (Supplementary Table 8) and 22.2% of Ck19+ liver cells in normal mouse livers (Fig. 2g and Supplementary Fig. 9), similar to the data from 5′-scRNA-seq analysis (Fig. 2d and Supplementary Table 6). Furthermore, RNAscope and multiplex IF data confirmed that these primary cells expressed CK19, DUOX2 and ACE2 (Fig. 2h, i). Moreover, primary Ck19+Duox2+Ace2+ cells also exhibited higher levels of Ck7 and Sox9 mRNA transcripts, the cholangiocyte specific-markers (Fig. 2j), defined as Duox2+Ace2+ cholangiocytes. Interestingly, all these human and mouse DUOX2+ACE2+ cholangiocytes had diameters of ≤10 μM (Fig. 2h, i), therefore belonging to small cholangiocytes[11]. Real-time qPCR analysis also indicated that the mRNA transcripts of cystic fibrosis transmembrane conductance regulator (Cftr) and the secretin receptor (Sctr), two typical markers of large cholangiocytes, were lower in mouse Duox2+Ace2+ small cholangiocytes than in the Duox2-Ace2- cholangiocytes (Supplementary Fig. 10a). Moreover, further analysis of DEGs revealed that DUOX2+ACE2+ cholangiocytes highly expressed the marker genes of small cholangiocytes (FOXA2 and HNF4A)[12], but expressed low levels of the marker genes of large cholangiocytes (SCTR/SR, CASP9 and IL2RB)[13,14] (Supplementary Fig. 10b). Together, these findings suggest that the identified DUOX2+ACE2+ small cholangiocytes may be targets for PBC.

### A selective decrease in the number of DUOX2+ACE2+ small cholangiocytes is significantly associated with impairment of bile secretion and the severity of PBC

Kyoto Encyclopedia of Genes and Genomes (KEGG) pathway analysis indicated that the bile secretion pathway was enriched in human DUOX2+ACE2+ small cholangiocytes compared to other clusters of cholangiocytes (Fig. 3a). Further analysis of DEGs revealed significantly increased expression of key genes regulating bicarbonate and/or chloride secretion into bile, including the transmembrane member 16A gene (TMEM16A/ANO1), the type 3 isoform of the inositol-1,4,5-triphosphate receptor (ITPR3), and the electrogenic sodium bicarbonate cotransporter (NBC/SLC4A4) (Supplementary Fig. 11). These observations suggest that DUOX2+ACE2+ small cholangiocytes may be crucial for bile secretion and their deficiency should impair bile secretion to promote cholestasis during the pathogenic process of PBC. Interestingly, RNAscope and multiplex IF analyses of liver sections displayed that the numbers of DUOX2+ACE2+ small cholangiocytes in small bile ducts were dramatically reduced in PBC patients, compared to control patients (Fig. 3b–e, Supplementary Figs. 12 and 13, and Supplementary Table 9). Furthermore, the decreased numbers of DUOX2+ACE2+ small

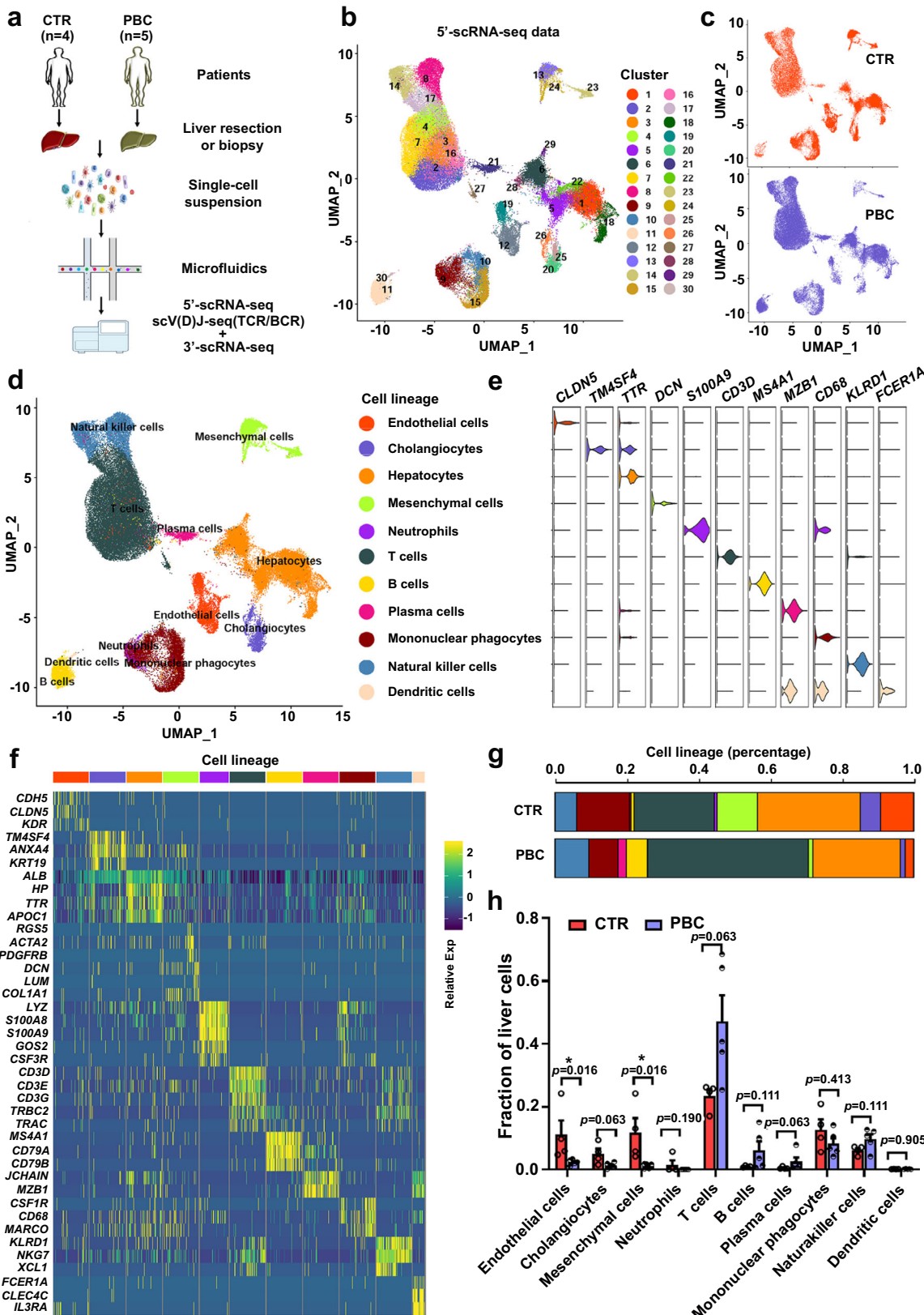

cholangiocytes were significantly associated with the severity of PBC, including higher Nakanuma stages, Ludwing stages, fibrosis scores, bile duct loss scores, and cholangitis activity grades (Fig. 3f). Notably, substantially reduced numbers of hepatic DUOX2⁺ACE2⁺ small cholangiocytes were observed in PBC patients with positive AMA-M2 (PBC-M2+) or negative AMA-M2 (PBC-M2−), but not in patients with obstructive cholestasis (OC), secondary sclerosing cholangitis (SSC), and nonalcoholic steatohepatitis (NASH) (Fig. 3g and Supplementary Table 9). These data indicated that the numbers of hepatic DUOX2⁺-ACE2⁺ small cholangiocytes are substantially decreased in PBC, and significantly associated with the impairment of bile secretion and the severity of PBC.

**Fig. 1 | Single-cell transcriptomic atlas of human liver cells. a** The strategies for preparing single liver cells for single-cell RNA sequencing (scRNA-seq). **b** UMAP analysis of 70,050 liver cells from four control (CTR) and five untreated primary biliary cholangitis (PBC) patients identified 30 distinct cell clusters. **c** Cluster distribution of cells in CTR livers (orange) and PBC livers (blue). **d** UMAP plot visualizing the distribution of identified 11 liver cell lineages. **e** Violin plot showing the expression levels of well-known representative enriched marker genes across 11 cell lineages, colored by cell lineage. **f** Heatmap displayed the relative expression level of marker genes in 11 cell lineages (top, color-coded by cell lineages), with exemplar genes labeled (right). Columns denote cells; rows denote genes. **g** The relative percentage of different cell lineages for each group, colored by cell lineage. CTR: $n = 4$ liver samples from CTR patients, PBC: $n = 5$ liver samples from PBC patients. **h** Comparision of the proportions of different cell lineages between CTR livers and PBC livers. CTR: $n = 4$ liver samples from CTR patients, PBC: $n = 5$ liver samples from PBC patients; Plotted: mean ± SEM; Statistics: two-tailed Mann−Whitney $U$ test, 95% confidence interval; *$p < 0.05$. Source data are provided as a Source Data file.

## CD27[+] memory B and plasma cells accumulate in the hepatic portal areas of patients with PBC and interact with DUOX2[+]-ACE2[+] small cholangiocytes

It is well known that PBC is caused by autoimmunity-mediated biliary injury[1–3]. To investigate which immune cell infiltrates were related to DUOX2[+]ACE2[+] small cholangiocyte injury, we analyzed serial frozen liver sections from a control and PBC patient using hematoxylin-eosin (HE) (Fig. 4a), spatial transcriptomics (ST) (Fig. 4b–e), and multiplex IF analyses (Fig. 4f and Supplementary Fig. 14). Our ST data revealed that transcriptomes were obtained from 3573 or 1365 spots with a median depth of 6302 or 3500 UMIs/spot and 1763 or 1454 genes/spot in the control or PBC liver section, respectively (Fig. 4b). Unbiased clustering of ST spots in liver sections exhibited the expression of genes mapping to liver cells in scRNA-seq (Fig. 4c). The proportions of B, plasma, and T cells increased in PBC liver sections, compared to the control liver sections (Fig. 4c). Hepatic portal ST data further indicated that B and plasma cell infiltrates significantly increased in the portal areas of the PBC liver, compared to that of the control liver (Fig. 4d, e). In contrast, the proportions of monocytes and NK cells significantly decreased in the portal areas of the PBC liver (Fig. 4e). There was no significant difference in the proportions of T, Kupffer and dendritic cells between the PBC and control livers (Fig. 4e). Multiplex IF analyses of serial frozen human liver sections exhibited that the numbers of DUOX2[+]-ACE2[+] small cholangiocytes (white color) markedly decreased while B (CD20[+], green color) and plasma cells (CD138[+], purple color) dramatically increased in the portal areas of the PBC liver, compared to the control liver (Fig. 4f and Supplementary Fig. 14). Further IF analyses of B cell subpopulation displayed that the numbers of CD27[+] memory B (CD20[+]IgD[−]CD27[+], orange color) cells were markedly higher in the portal area of the PBC liver than that of the control liver (Fig. 4f and Supplementary Fig. 14). Analyses of B cell receptor (BCR) profiling revealed that plasma cells displayed an increased clonal expansion in PBC patients compared with that of control patients (Fig. 4g).

Next, we analyzed the potential interaction between DUOX2[+]-ACE2[+] small cholangiocytes and immune cells, using the CellPhoneDB based on ligand-receptor interactions[15]. This analysis revealed significant interactions between ligands and receptors expressed by DUOX2[+]ACE2[+] small cholangiocytes and immune cells, including B and plasma cells (Fig. 5a–c). Furthermore, there were also significant ligand-receptor interactions between DUOX2[+]ACE2[+] small cholangiocytes (or immune cells) and cholangiocytes (1) or (2) (Fig. 5d–f and Supplementary Fig. 15). These findings indicated that the immune response was associated with a selective decrease in the numbers of liver DUOX2[+]ACE2[+] small cholangiocytes and might also affect other types of cholangiocytes by cell-cell interactions during the process of PBC pathogenesis.

## Hepatic pIgR is highly expressed in DUOX2[+]ACE2[+] small cholangiocytes and the levels of serum anti-pIgR antibodies are significantly elevated in PBC patients

To investigate which and how autoantibodies produced by plasma cells are associated with DUOX2[+]ACE2[+] small cholangiocyte injury in the liver, we performed proteomics analyses of serum samples from control subjects and PBC patients (Fig. 6a, b and Supplementary Table 10). Among 2805 quantified proteins, 50 serum proteins increased and 87 decreased in PBC patients compared to that of control subjects (Fig. 6c). The volcano plot of differential proteins exhibited that the levels of serum pIgR in PBC patients were substantially higher than that of control patients (Fig. 6d, e and Supplementary Table 11). Similarly, the levels of serum anti-pIgR antibodies were also significantly elevated in PBC patients, including those patients with negative AMA-M2 (Fig. 6f and Supplementary Table 12). Interestingly, *pIgR* was mainly expressed in cholangiocytes from human livers (Fig. 6g), consistent with previous observations[9,10]. The scRNA-seq, RT-qPCR, and multiplex IF analyses of human and/or mouse liver cells further revealed that the levels of pIgR mRNA transcripts and proteins were dramatically higher in DUOX2[+]ACE2[+] small cholangiocytes than that in DUOX2[−]ACE2[−] cholangiocytes and other types of liver cells (Fig. 6h–j). Most importantly, multiplex IF displayed that the pIgR protein was mainly detected in DUOX2[+]ACE2[+] small cholangiocytes in the control livers, whereas its expression was markedly reduced in the PBC liver, along with the reduced numbers of DUOX2[+]ACE2[+] small cholangiocytes (Fig. 6k, l). However, reduced number of DUOX2[+]ACE2[+] small cholangiocytes was not detected in SSC, OC and NASH livers (Supplementary Fig. 16). Together, our findings suggest that pIgR-specific autoantibodies may be associated with DUOX2[+]ACE2[+] small cholangiocyte injury in the pathogenic process of human PBC.

## Discussion

Here, we have identified a unique type of DUOX2[+]ACE2[+] small cholangiocytes in human and mouse livers, whose selective decrease was associated with the development and progression of PBC. The current study highlights four major findings: (1) Unique DUOX2[+]ACE2[+] small cholangiocytes were identified in human and mouse livers (Fig. 2); (2) A selective decrease in the numbers of hepatic DUOX2[+]ACE2[+] small cholangiocytes was observed in PBC patients, but not in OC, SSC, and NASH patients and their selective loss was associated with the severity of PBC and cholestasis (Fig. 3); (3) Hepatic pIgR was highly expressed in DUOX2[+]ACE2[+] small cholangiocytes and the levels of serum anti-pIgR autoantibodies were markedly elevated in PBC patients, regardless of positive and negative AMA-M2, but not in OC patients (Fig. 6 and Supplementary Fig. 16); and (4) CD27[+] memory B and plasma cells accumulated in portal areas of the liver in PBC patients where there were significant ligand-receptor interactions between DUOX2[+]ACE2[+] small cholangiocytes and immune cells, including B and plasma cells (Figs. 4 and 5). These findings support an immunological mechanism underlying the pathogenesis of PBC and may provide an explanation as to why small bile ducts are selectively targeted in PBC. Our findings also suggest that DUOX2[+]ACE2[+] small cholangiocytes and serum anti-pIgR autoantibody levels may be specific biomarkers for the diagnosis and evaluation of PBC, including PBC patients with negative AMAs. Conceivably, preservation of DUOX2[+]ACE2[+] small cholangiocytes and targeting anti-pIgR autoantibodies may be valuable strategies for future therapeutic interventions for PBC.

PBC is a chronic autoimmune liver disease with a female predominance[1–3]. Previous studies suggest that the presence of AMAs results from the loss of immune tolerance to PDC-E2 and that autoreactive T cell immunity in PBC is the cause of injury to BECs[4–6]. However, some PBC patients are negative for AMA-M2[1–3], suggesting

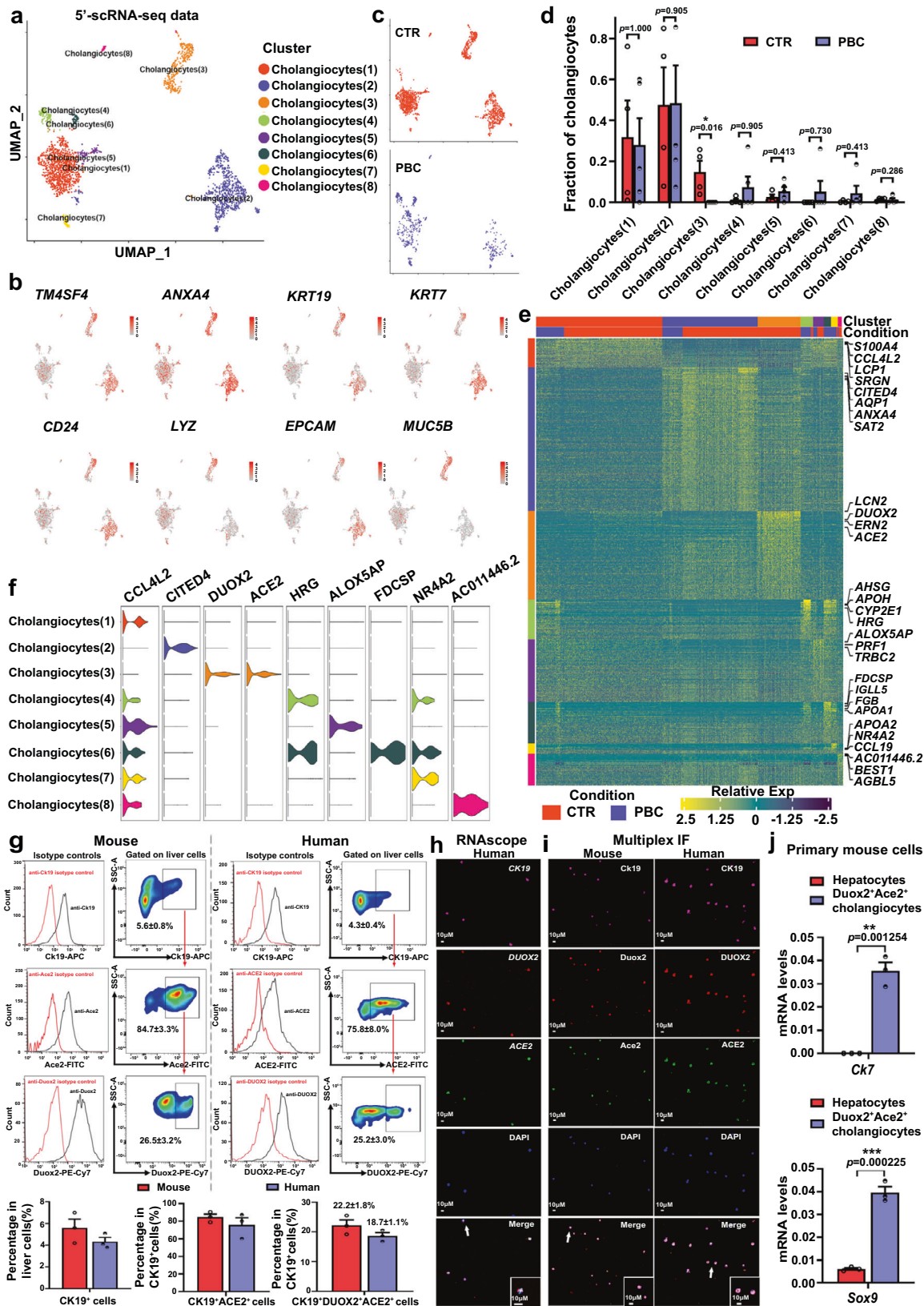

that PBC may not always be dependent on AMA-specific humoral responses. Until now, it is unclear why the pathogenic autoimmune response selectively targets small BECs or how patients with negative AMAs develop PBC. Here, we identified a unique type of hepatic DUOX2⁺ACE2⁺ small cholangiocytes as the pathogenic target of PBC, because these cells express high levels of pIgR, and significantly

elevated levels of serum anti-pIgR autoantibodies exist in PBC patients (Fig. 6). It is worth to mention that CD27⁺ memory B and plasma cells markedly accumulated in the hepatic portal areas of PBC patients (Fig. 4), likely producing anti-pIgR autoantibodies. Proteomic analysis of serum samples indicated that the intestinal immune network for IgA production pathway was enriched in PBC patients (Supplementary

**Fig. 2 | Identification of unique DUOX2⁺ACE2⁺ small cholangiocytes in human and mouse livers. a** UMAP analysis of 2,209 cholangiocytes from four control (CTR) and five primary biliary cholangitis (PBC) livers identified eight distinct clusters of cholangiocyte (1) to (8). **b** Canonical marker genes were highly expressed in different clusters of cholangiocytes in human livers. **c** Cluster distribution of cells in control livers (orange) and PBC livers (blue). **d** The proportions of cholangiocyte subpopulations in control and PBC livers. CTR: $n = 4$ liver samples from CTR patients, PBC: $n = 5$ liver samples from PBC patients; Plotted: mean ± SEM; Statistics: two-tailed Mann-Whitney $U$ test, 95% confidence interval; *$p < 0.05$. **e** Heatmap displayed the relative expression levels of marker genes in different clusters of cholangiocytes (top, color-coded by cluster and condition), with exemplar genes labeled (right). Columns denote cells; rows denote genes. **f** Violin plots showing the expression levels of enriched marker genes across distinct cholangiocyte subpopulations. **g** Flow cytometry analysis and cell sorting of primary CK19⁺DUOX2⁺ACE2⁺ cells in the livers of three wild-type mice and three control patients. Mouse: $n = 3$ mouse liver samples, Human: $n = 3$ human liver samples; Plotted: mean ± SEM. **h** Representative photomicrographs of sorted primary human CK19⁺DUOX2⁺ACE2⁺ cells that had been analyzed using RNAscope with probes for *CK19* (purple), *DUOX2* (red) and *ACE2* (green). Scale bars: 10 μm. $n = 3$ independent experiments. **i** Representative photomicrographs of sorted primary mouse Ck19⁺Duox2⁺Ace2⁺ cells and human CK19⁺DUOX2⁺ACE2⁺ cells analyzed using multiplex immunofluorescence (IF) with antibodies against CK19 (purple), DUOX2 (red) and ACE2 (green). Scale bars: 10 μm. $n = 3$ independent experiments. **j** Specific cholangiocyte markers, *Ck7* and *Sox9*, were highly expressed in primary mouse Ck19⁺Duox2⁺Ace2⁺ cells, but not in primary mouse hepatocytes. Plotted: mean ± SEM. $n = 3$ independent experiments; Statistics: two-tailed independent-sample Student's $t$ test, 95% confidence interval; **$p < 0.01$, ***$p < 0.001$. Source data are provided as a Source Data file.

Fig. 17), potentially increasing the levels of serum IgA. Moreover, there were significant ligand-receptor interactions between DUOX2⁺ACE2⁺ small cholangiocytes and immune cells (Fig. 5a–c). Accordingly, we propose an immunological mechanism underlying the pathogenesis of PBC (Fig. 7). Although it is not known how anti-pIgR autoantibody is generated, it is possible that IgA transcytosis mediated by the pIgR in DUOX2⁺ACE2⁺ small cholangiocytes and gut cells may accidentally present pIgR as antigen to immune cells. Subsequently, its loss of tolerance induces pIgR-specific humoral and T cell autoimmunity that produces anti-pIgR autoantibodies and recruits pIgR antigen- specific CD27⁺ memory B and plasma cells to the hepatic portal tracts. The elevated levels of anti-pIgR autoantibodies, together with other antigen-specific autoantibodies, further damage DUOX2⁺ACE2⁺ small cholangiocytes in the small bile ducts, impairing bile secretion, and contribute to the development of PBC, even if patients fail to develop AMA-M2 antibodies. In addition, since intestine cells also express pIgR[16], one may wonder why PBC patients does not have gut injury. We speculate that the bile duct epithelium is facing a much more hostile environment than intestinal cells, since the bile contains millimolar concentrations of bile acids in bile lumen. However, a recent case report indicates that gut injury is also seen not in all but in some PBC patients, who also have inflammatory bowel disease (IBD)[17]. This would support the idea that autoantibody against pIgR plays a role in PBC and some patients with IBD. However, future studies will need to test this hypothesis.

DUOX2⁺ACE2⁺ small cholangiocytes expressed high levels of key genes regulating bile secretion such as *ANO1, ITPR3, SLC4A4* (Supplementary Fig. 11). A selective decrease in the numbers of DUOX2⁺ACE2⁺ small cholangiocytes in PBC patients significantly promoted the progression of PBC and cholestasis (Fig. 3). These findings support the notion that the function of DUOX2⁺ACE2⁺ small cholangiocytes is crucial for bile secretion and the pathogenic targets of PBC. A recent study reports that ACE2 expression is significantly higher in cholangiocytes than in hepatocytes[18]. Interestingly, ACE2 is a receptor of severe acute respiratory distress syndrome coronavirus-2 (SARS-CoV-2)[19], which can cause persistent cholestasis in some survivors of severe COVID-19 patients[20,21]. It is possible that SARS-CoV-2 infection may damage DUOX2⁺ACE2⁺ small cholangiocytes, a similar pathogenic process as for PBC, supporting the notion that SARS-CoV-2 infection might promote the development of PBC[22]. However, more relevant studies would need to investigate whether SARS-CoV-2 infection is related to PBC progression.

In summary, we report a unique type of DUOX2⁺ACE2⁺ small cholangiocytes in the liver, and show that these cells are crucial for bile secretion and are the pathogenic targets of PBC. Furthermore, we discover that the hepatic pIgR is highly expressed in DUOX2⁺ACE2⁺ small cholangiocytes and that the levels of anti-pIgR antibodies are increased in PBC patients, accompanied by increased numbers of CD27⁺ memory B and plasma cells in hepatic portal tracts. Our data suggest that a pIgR-specific humoral responses may selectively destroy DUOX2⁺ACE2⁺ small cholangiocytes to impair bile secretion, promoting the pathogenic process of PBC. Thus, our findings may have uncovered the pathogenic targets of PBC and identified an immunological mechanism underlying the pathogenesis of PBC including the explanation for why some patients develop PBC, independent of the presence of AMAs. DUOX2⁺ACE2⁺ small cholangiocytes and serum levels of anti-pIgR autoantibodies may also be specific biomarkers for the diagnosis and evaluation of PBC. Conceivably, preservation of DUOX2⁺ACE2⁺ small cholangiocytes and targeting anti-pIgR autoantibodies may be therapeutic strategies for intervention of PBC.

## Methods
### Human serum and liver samples
This study was performed, according to the Declaration of Helsinki (2013) of the World Medical Association. The study protocol was approved by the Ethics Committees of the First Affiliated Hospital of Third Military Medical University and the Xiangya Hospital of Central South University. Export of human genetic information and materials was approved by China's Ministry of Science and Technology (approval number 2022BAT1359). A corresponding written informed consent was obtained from individual subjects. For scRNA-seq and ST analyses, biopsied liver samples were obtained from five PBC patients with no history of ursodeoxycholic acid (UDCA) treatment (Supplementary Table 1). These PBC patients were diagnosed by an elevated serum alkaline phosphatase (ALP) level and positive anti-mitochondrial antibodies (AMAs). Their liver tissues were biopsied to histologically exclude autoimmune hepatitis and other liver diseases (Supplementary Table 2 and Supplementary Fig. 1)[23]. Control surgical liver samples were obtained from four patients, who had no evidence of PBC, cholestasis, viral hepatitis, and autoimmune hepatitis (Supplementary Table 1). Three surgical liver samples were obtained from control patients for fluorescence-activated cell sorting (FACS) analysis of cholangiocytes (Supplementary Table 8). Moreover, surgical or biopsied liver samples from 16 control patients, 1 nonalcoholic steatohepatitis (NASH), 1 obstructive cholestasis (OC), 1 secondary sclerosing cholangitis (SSC) and 18 PBC patients, and an intestinal sample form 1 control patient were used for multiplex IF, RNAscope and liver histological assessments (Supplementary Table 9). The histological sections were evaluated by an experienced hepatopathologist in a blinded fashion and the degrees of fibrosis, bile duct loss, cholangitis and hepatitis activity in each sample were scored using the Ludwig and Nakanuma systems[24,25]. Blood samples from six healthy volunteers and 10 PBC patients were used for proteomics (Supplementary Table 10). In addition, serum samples from 12 control subjects, 17 PBC patients, and 15 OC patients were used to quantify the levels of serum anti-pIgR antibodies by the ELISA assay (Supplementary Table 12).

### Single cell preparation from human livers for scRNA-seq analysis
The surgical or biopsied control and PBC liver tissues in the GEXSCOPETM Tissue Preservation Solution were washed with

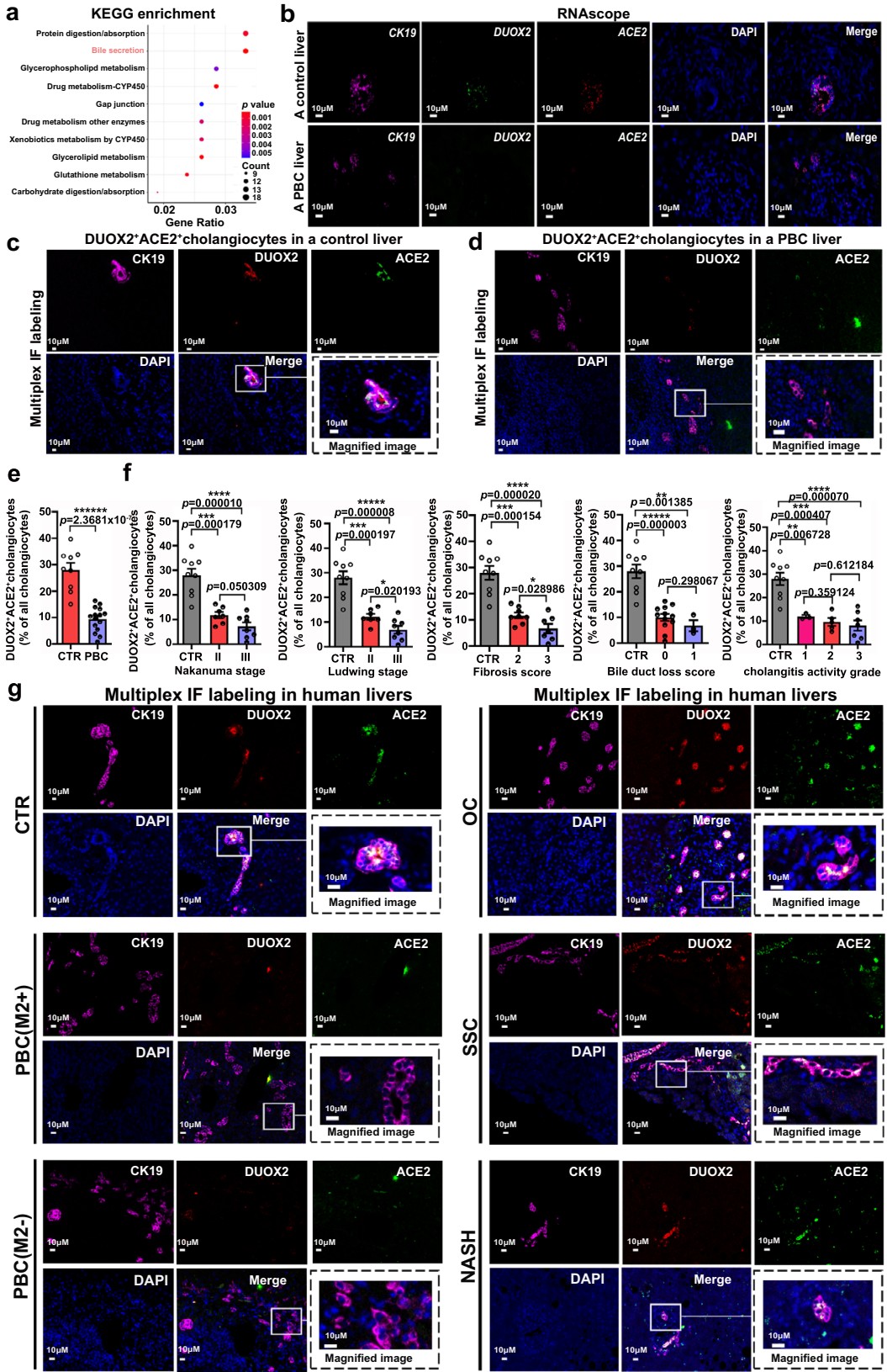

phosphate-buffered saline (PBS, Hycolone, Cat#SA30256.01) and minced into small pieces. The tissue pieces were digested in GEXSCOPE® Tissue Dissociation Solution (Singleron, Cat#1020012) at 37 °C for 20 min with consistent agitation and filtered through 70 μm sterile cell strainer (Falcon, Cat#352340), followed by centrifugation. After lysis of red blood cells (RBC), the remaining cells were examined for their viability (a cell preparation with >90% viability was used for subsequent experiments). A similar protocol was used for preparing mouse liver tissue single-cell suspension except for using 45% Percoll buffer (cytiva, Cat#17089109) to wash the cell pellets once before RBC lysis.

**Fig. 3 | Functional and clinical significance of DUOX2⁺ACE2⁺ small cholangio-cytes in PBC. a** Kyoto Encyclopedia of Genes and Genomes (KEGG) pathway analysis indicated that the bile secretion pathway was enriched in human DUOX2⁺ACE2⁺ small cholangiocytes compared to other cholangiocyte subpopulations. The x axis represents the proportion of genes enriched in this pathway accounting for all differentially expressed genes of DUOX2⁺ACE2⁺ small cholangiocytes. Circle size denotes the number of genes enriched in one pathway; color (red, high; blue, low) denotes the singificance of one pathway. Statistics: two-tailed Hyper Geometric Test, 95% confidence interval. **b** Representative photomicrographs of control (CTR) and primary biliary cholangitis (PBC) liver tissue sections analyzed by RNAscope with probes for *CK19* (purple), *DUOX2* (green) and *ACE2* (red). Scale bars: 10 µm. *n* = 3 independent experiments. **c, d** Representative photomicrographs of CTR (**c**) and PBC (**d**) liver tissue sections stained by multiplex immunofluorescence (IF) with antibodies against CK19 (purple), DUOX2 (red) and ACE2 (green). Scale bars: 10 µm. *n* = 3 independent experiments. **e** The proportion of DUOX2⁺ACE2⁺ cholangiocytes in all cholangiocytes for CTR and PBC patients.

CTR: *n* = 9 liver samples from CTR patients, PBC: *n* = 15 liver samples from PBC patients; Plotted: mean ± SEM; Statistics: two-tailed independent-sample Student's *t*-test, 95% confidence interval; *******p* < 0.000001. **f** The decrease in the number of DUOX2⁺ACE2⁺ small cholangiocytes was significantly associated with higher Nakanuma and Ludwing stages, fibrosis and bile duct loss scores, and cholangitis activity grades. CTR: *n* = 9 liver samples from CTR patients, PBC: *n* = 15 liver samples from PBC patients; Plotted: mean ± SEM; Statistics: two-tailed independent-sample Student's *t*-test, 95% confidence interval; *$p < 0.05$,**$p < 0.01$, ***$p < 0.001$, ****$p < 0.0001$, *****$p < 0.00001$, ******$p < 0.00001$. **g** Representative photomicrographs of the human liver tissue sections from CTR, PBC with positive AMA-M2 or negative AMA-M2, obstructive cholestasis (OC), secondary sclerosing cholangitis (SSC), and non-alcoholic steatohepatitis (NASH) patients after multiplex IF with antibodies against CK19 (purple), DUOX2 (red) and ACE2 (green). Scale bars: 10 µm. *n* = 3 independent experiments. Source data are provided as a Source Data file.

## Droplet-based scRNA-seq

Individual single-cell suspensions ($1 \times 10^5$ cells/ml) from five PBC and four control liver tissues were barcoded to generate scRNA-seq libraries using the Chromium Single Cell 3′ and 5′ Library, Gel Bead & Multiplex Kit (10x Genomics), per the manufacturer's instruction. Briefly, individual cells were partitioned into Gel Beads in Emulsion for cell lysis and barcoded RNA reverse transcription in the ChromiumTM Controller instrument. Similarly, the cDNAs from individual lymphocyte samples were used for generation of T cell receptor (TCR)- and B cell receptor (BCR)-enriched libraries using the Chromium Single Cell V(D)J Enrichment kit. All the libraries were subjected to next-generation sequencing (NSG) on Illumina Novaseq 6000 with 150 bp paired end reads.

## scRNA-seq data pre-processing

The raw reads from scRNA-seq were processed to generate gene expression matrixes using an internal pipeline. Briefly, the raw reads were firstly analyzed using fastQC (version 0.11.4, https://www.bioinformatics.babraham.ac.uk/projects/fastqc/)[26] and fastp (version 1, https://github.com/OpenGene/fastp)[27] to remove low quality of reads, and with cutadapt (version 1.17, https://github.com/marcelm/cutadapt/)[28] to eliminate poly-A tail and adapter sequences. After extraction of cell barcode and unique molecular identifier (UMI), the reads were mapped to the reference genome GRCh38 (nsemble version 92 annotation) using STAR (version 2.5.3a, https://github.com/alexdobin/STAR)[29]. The UMIs and genes of each cell were counted using the featureCounts (version 1.6.2, https://subread.sourceforge.net/)[30] to generate expression matrix files for subsequent analysis.

## Quality control, dimension-reduction and clustering for scRNA-seq data

Individual cells were filtered with a UMI cutoff value of <30,000 and gene value between 200 and 5000, and the cells with mitochondrial gene content >50% were deleted (Supplementary Table 3). Next, the functions of genes in individual cells were subjected to dimension-reduction process and clustered using the Seurat (version 3.1.2, http://satijalab.org/seurat/)[31]. Subsequently, the functions of genes in individual cells were normalized and scaled their expression using NormalizeData() and ScaleData(). The top 2000 variable genes in individual cells were selected using the FindVariableFeautres function for principal component analysis (PCA). The top 20 principal components were used to separate cells into multiple clusters with FindClusters(). The potential batch effect between samples was analyzed by Harmony (version 1.0, https://github.com/immunogenomics/harmony)[32]. The agreement between samples was examined by calculating the Pearson correlation coefficients between all pairwise comparisons of samples per lineage. Finally, the cells in a two-dimensional space were visualized using UMAP algorithm.

## Differentially expressed genes (DEGs) analysis and cell annotation using scRNA-seq data

The DEGs were identified as a gene expression in >10% of cells in a cluster with an average log (fold change, FC) value of >0.25 between cells, using the FindMarkers() function in the Seurat (version 3.1.2) based on Wilcox likelihood-ratio test. Individual cells in each cluster were annotated, according to the expression of canonical marker genes in the DEGs using literature knowledges (Supplementary Tables 4 and 13). The expression of markers of each cell type was visualized in heatmaps/dot plots/violin plots using the DoHeatmap()/DotPlot()/Vlnplot() function in the Seurat (version 3.1.2). The potential doublet cells were identified by their expressing markers, and removed manually.

## Pathway enrichment analysis

The potential functions of DEGs were analyzed by the Gene Ontology (GO) and Kyoto Encyclopedia of Genes and Genomes (KEGG) using the clusterProfiler package (version 4.4.4, http://bioconductor.org/packages/release/bioc/html/clusterProfiler.html) in the R software (version 3.6.3, https://www.r-project.org/)[33]. Pathways with a *p*_adj value of <0.05 were considered as significantly enriched. The GO gene sets, including molecular function (MF), biological process (BP), and cellular component (CC) categories, were used as reference.

## TCR and BCR analysis

The V(D)J sequencing data were analyzed by the Cellranger (version 3.1.0, https://github.com/10XGenomics/cellranger). After removing cells with only a single chain (light or heavy chain for BCR, and α chain or β chain for TCR), the remaining cells were annotated to V(D)J data using the 5′ single-cell sequencing data. Subsequently, the frequency of different clonotypes of cells were amounted, according to the samples in different cell lineages. All clonotypes were divided into three categories, based on the number of cells they presented. The first type was a clonotype in only one cell, the second was a clonotype in two cells, and the third was a clonotype in three or more cells. The proportion of each clonotype in each cell subtype was calculated.

## Cell−cell interaction analysis

The cell-cell interactions were analyzed using CellPhoneDB (version 2.1.0, https://github.com/Teichlab/cellphonedb), based on the receptor−ligand interactions between two types of cells[15]. Cluster labels of all cells were randomly permuted for 1000 times to calculate the null distribution of average ligand-receptor expression levels of the interacting clusters. The expression levels of individual ligands or receptors were analyzed for a cutoff value based on the average log gene expression distribution for all genes across all the cell types. The significant cell-cell interactions were defined as *p* < 0.05 and average log expression >0.1, which were visualized with the Circlize package

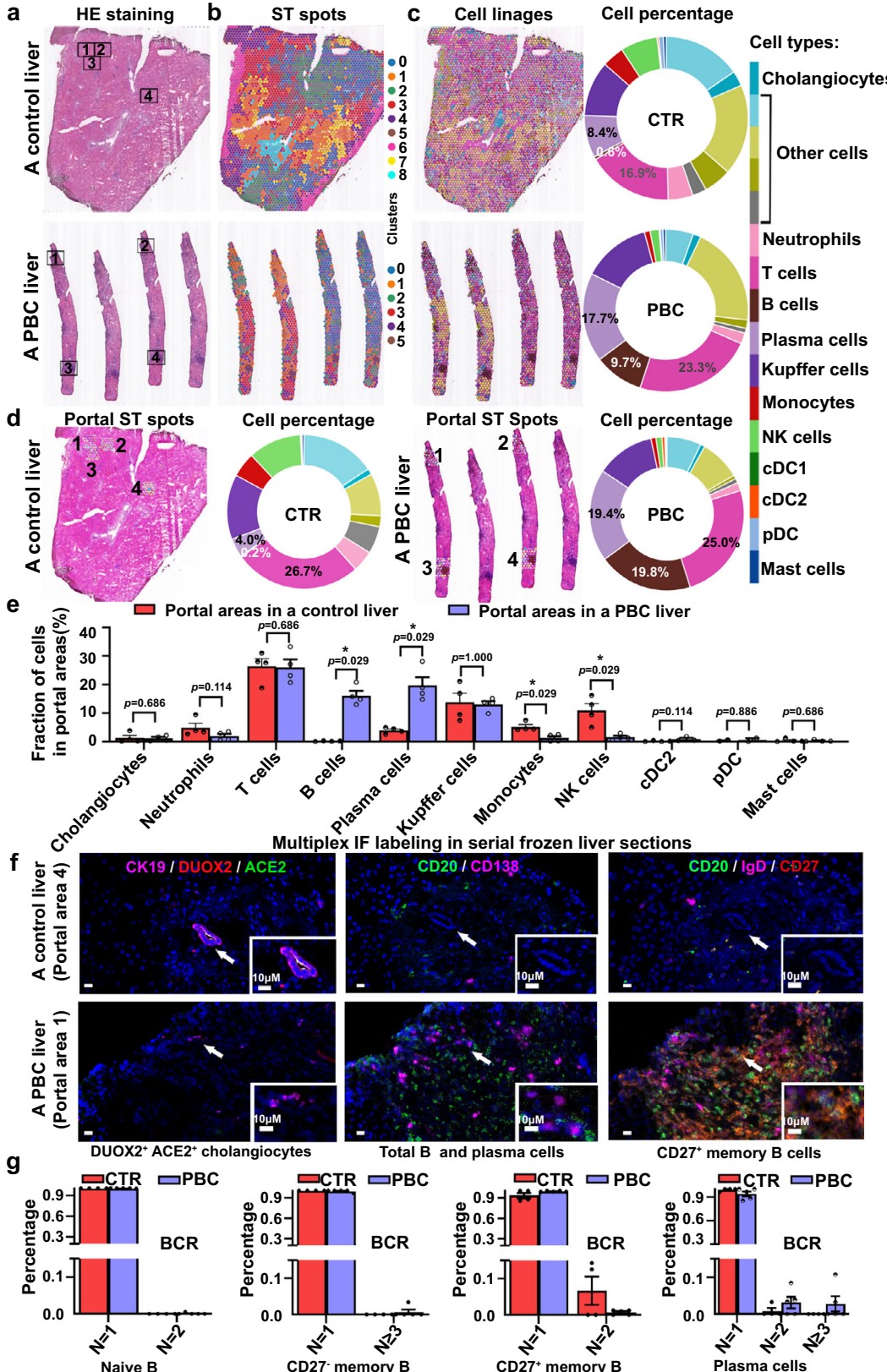

**a** HE staining **b** ST spots **c** Cell linages Cell percentage

**d** Portal ST spots Cell percentage Portal ST Spots Cell percentage

**e** Portal areas in a control liver / Portal areas in a PBC liver

**f** Multiplex IF labeling in serial frozen liver sections

CK19 / DUOX2 / ACE2    CD20 / CD138    CD20 / IgD / CD27

DUOX2⁺ ACE2⁺ cholangiocytes    Total B and plasma cells    CD27⁺ memory B cells

**g** CTR PBC

Naive B    CD27⁻ memory B    CD27⁺ memory B    Plasma cells

(version 0.4.15, https://cran.r-project.org/web/packages/circlize/) in the R software (version 3.6.3).

## Spatial Transcriptomics (ST) of human livers

The ST of fresh liver samples was performed according to the manufacturer's instructions (10× Genomics). In brief, fresh liver samples from a control subject and a PBC patient were snap-frozen and embedded in Optimal Cutting Temperature (OCT). The liver cryosections (10 μm) on the Capture Areas of the Visium Spatial Gene Expression slides were fixed in methanol and stained with Hematoxylin and eosin (HE) (DAKO, Cat#S330930-2). The stained liver sections were imaged for subsequent mapping the gene expression patterns.

**Fig. 4 | CD27⁺ memory B and plasma cells accumulate in the hepatic portal areas of PBC patients. a** Images of hematoxylin-eosin (HE) staining in liver sections from a control (CTR) subject and a primary biliary cholangitis (PBC) patient. Four typical portal areas were randomly selected from each section for further analyses. **b** Unbiased clustering of spatial transcriptomic (ST) spots in the CTR and the PBC liver sections, which identified 9 and 6 distinct clusters, respectively. **c** The distribution of cell linages and their proportions in the CTR and PBC liver sections. NK cells natural killer cells, cDC1 conventional type 1 dendritic cells, cDC2 conventional type 2 dendritic cells, pDC plasmacytoid dendritic cells. **d** Spatial distribution of annotated cell linages and their proportions in the selected 4 portal areas of the CTR and PBC liver sections. **e** The proportions of different types of cells in the portal areas of the CTR and PBC liver sections. CTR: $n = 4$, PBC: $n = 4$; Plotted: mean

± SEM; Statistics: two-tailed Mann–Whitney $U$ test, 95% confidence interval; *$p < 0.05$. **f** Representative multiplex immunofluorescence (IF) photomicrographs of DUOX2⁺ACE2⁺ small cholangiocytes (CK19⁺ACE2⁺DUOX2⁺, white), B cells (CD20⁺, green), plasma cells (CD138⁺, purple), and CD27⁺ memory B cells (CD20⁺IgD⁻CD27⁺, orange) in the portal areas of serial frozen liver sections of the CTR (Portal area 4) and PBC liver (Portal area 1). Scale bars: 10 μm. $n = 3$ independent experiments. **g** The distribution of B-cell receptor (BCR) clonotypes in B cell subtypes of the CTR and PBC groups. The N represents the number of cells for a clonotype; $N = 1$: a clonotype only in one cell; $N = 2$: a clonotype in two cells; and $N = 3$: a clonotype in ≥3 cells. The $y$ axis represents the percentage of each clonotype in total clonotypes. CTR: $n = 4$ liver samples from CTR patients, PBC: $n = 5$ liver samples from PBC patients; Plotted: mean ± SEM. Source data are provided as a Source Data file.

The polyadenylated mRNAs released from the overlying cells were captured by the primers on the spots. The permeabilized sections were treated with RT Master Mix containing reverse transcription reagents to generate partially barcoded, full-length cDNA on the slide. The spatially barcoded, full-length cDNAs were amplified by PCR to construct the libraries. After enzymatic fragmentation and size selection, the cDNAs in the libraries were sequenced on the Illumina Novaseq 6000.

The ST data from two 10X Visium capture areas (5000 barcoded spots/area) were firstly log-normalized using the Seurat (version 3.1.2, https://satijalab.org/seurat/). During this process, feature counts for each spot were divided by the total counts for that cell, multiplied by the scale.factor(1e6), and then log-transformed using log1p. Next, the top variable genes were screened using the Seurat package and the FindVariableFeatures function. The top 2000 highly variable genes from each dataset were selected using the "vst" method. The distribution of top highly variable gene RNAs in each spot was analyzed by RunPCA and they were further reduced the dimensions to the top 20 principal components, followed by scaling and centering features in the dataset (ScaleData). The data on reduced UMAP dimensions were visualized by RunUMAP. The spots on PCA space were clustered using the FindNeighbours and FindClusters with the Shared Nearest Neighbor (SNN) algorithm. The ST data were further analyzed using stLearn (version 0.3.1, https://github.com/BiomedicalMachineLearning/stLearn), based on the spatial distance, tissue morphology and gene expression and images, followed by deconvoluting ST data by Seurat (version 3.1.2) using SCTransform and RunPCA. The anchors in each cell type in scRNA-seq and spatial RNA-seq data for potential integration were identified using the FindTransferAnchors. The underlying composition of each type of cells was predicted to deconvolute each of the spatial voxels using TransferData method. Subsequently, a table containing the proportions of individual cell types in spatial RNAseq data was achieved, which was then classified and visualized in each cluster using stLearn (version 0.3.1)[34].

## Cell preparation, flow cytometry analysis and sorting

Single liver cells were prepared through the same workflow of preparing single-cell suspension for scRNA-seq mentioned above from three control patients (two male and one female) and three wild-type (WT) 7-week-old male C57BL/6J mice (Gempharmatech, Jiangsu, China), and stained with anti-DUOX2 antibody or isotype control for 30 min at 4 °C, followed by staining with PE/CY7-anti-human/mouse IgG2a heavy chain antibodies for 30 min at 4 °C. After being washed, the cells were stained with both anti-CK19 Alexa Fluor® 647 and anti-ACE2 CoraLite® 488 antibodies or isotype controls for 30 min at 4 °C and nuclear stained with DAPI (1:1000). The frequency of CK19⁺DUOX2⁺ACE2⁺ cells and CK19⁺DUOX2⁻ACE2⁻ cells was analyzed by flow cytometry on a flow cytometer (BD FACSCanto II, BD Biosciences). All flow cytometry data were analyzed using the FlowJo software (version 10.6.2). The CK19⁺DUOX2⁺ACE2⁺ cells and CK19⁺DUOX2⁻ACE2⁻ cells were sorted for preparing total RNA extraction or microscope cell analysis. The detailed information on primary

or secondary antibodies is described in Supplementary Table 14. All mouse-related experiments were conducted according to the protocols approved by the Institutional Animal Care and Use Committee of the First Affiliated Hospital of Third Military Medical University (AMUWEC20211648).

## Preparation of microscope cell slides

The collected cholangiocytes in DMEM-F12 medium (HyClone, Cat#SH30023.01) were cultured onto collage type I cellware 22 mm round overslips (CORNING, Cat#354089) at 37 °C in a 5% CO₂ incubator overnight. After being washed with PBS for three times, the cells were fixed with 4% paraformaldehyde for 20 min, and stored at −80 °C until used.

## Multiplex immunofluorescence (IF) staining

Primary antibodies against CK19, CK7, ACE2, DUOX2, CD138, CD20, IgD, CD27, or pIgR were used to perform multiplex IF staining with the multiplex immunohistochemistry/immunofluorescence staining kits (absin, Shanghai, China, Cat# abs50030 and abs50014) according to the manufacturer's instructions (Supplementary Table 14). The specificity of CK19, ACE2 and DUOX2 antibodies was validated in the liver or intestinal tissue sections using a negative IgG control (Supplementary Fig. 18). The paraffin-embedded or frozen liver sections (4 μm) as well as liver cell slides were blocked with 8% goat serum (absin, Cat#abs933) in PBS for 10 min, and incubated at room temperature (RT) for 60 min sequentially with the primary antibodies. After being washed, the sections were probed with HRP-conjugated anti-rabbit or anti-mouse IgG at RT for 10 min and reacted with fluorophore-conjugated tyramine molecules (PPD 520, PPD 570, PPD 650 or PPD700) for 10 min. At the end of each cycle of staining, the primary and secondary antibodies were washed with Tris/EDTA buffer (PH 9.0) or eluent solution (absin, Cat#abs994), respectively. Subsequently, all sections were nuclear stained with DAPI for 5 min. The fluorescent signals were photo-imaged and merged using a laser scanning confocal microscope (ZEISS, LSM880) and ZEN software (version 2012).

## RNAscope in situ hybridization

The RNAscope assay was used to simultaneously measure human CK19, DUOX2 and ACE2 expression in fresh liver tissue sections or cell slides. RNAscope® Multiplex Fluorescent Reagent Kit v2 (Cat#323100), customized double Z oligonucleotide probes including RNAscope® Probe-Hs-DUOX2 (Cat#561361), RNAscope® Probe-Hs-ACE2-C2 (Cat#848151-C2), RNAscope® Probe-Hs-KRT19-C3 (Cat#310221-C3), RNAscope® 3-plex Positive Control Probe-Hs (Cat#320861) and RNAscope® 3-plex Negative Control Probe (Cat#320871), and opal fluorescent dye including Opal520 (Cat#ASOP520), Opal570 (Cat#ASOP570) and Opal690 (Cat#ASOP690), were purchased from Advanced Cell Diagnostics (ACD, Hayward, CA, USA). Liver tissue or liver cell preparation, pretreatment, target probe hybridization, hybridization signal amplification, and probe signal marking were performed according to the commercial kit's instructions[35]. The fluorescent signals were photo-imaged and merged using a laser

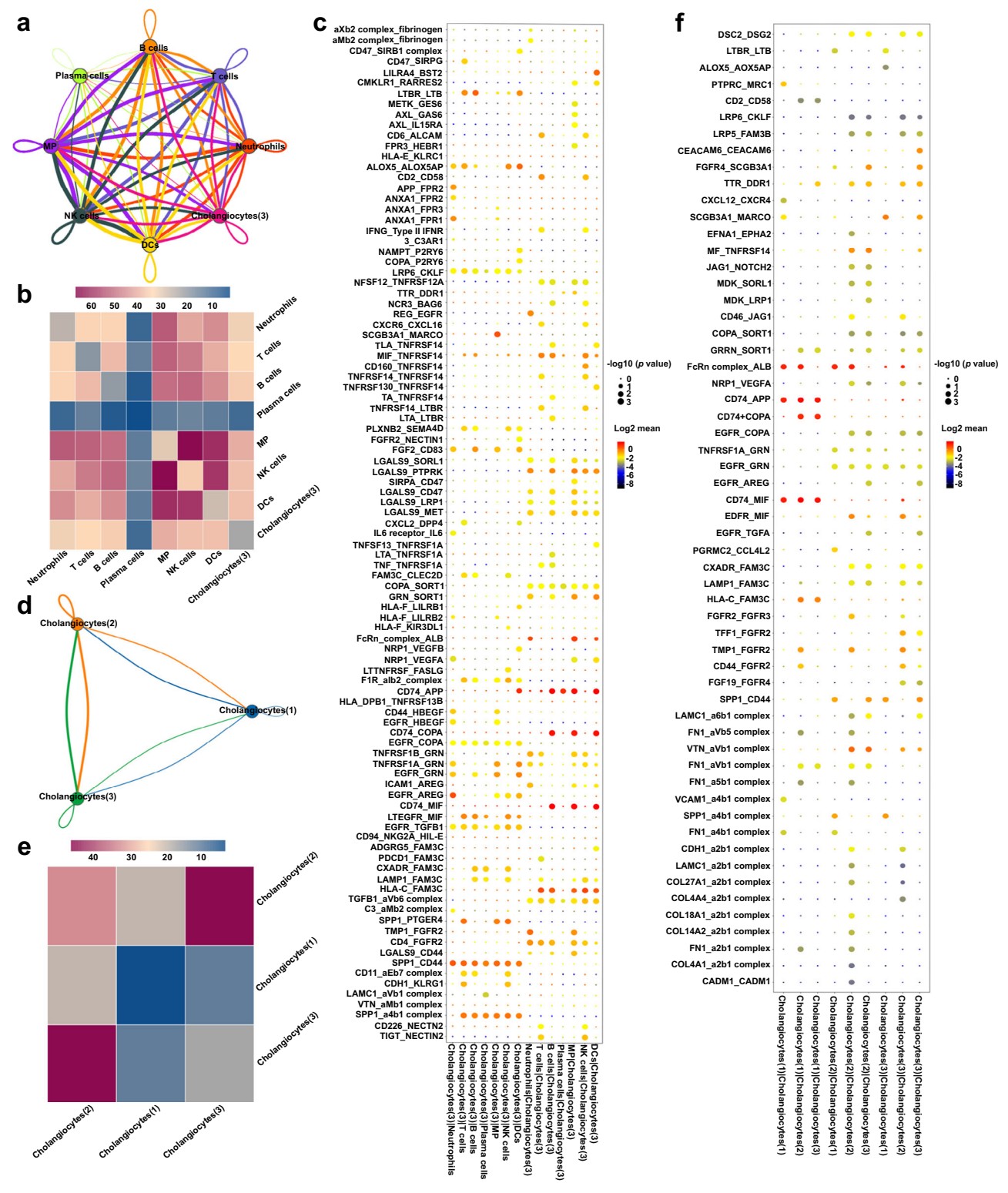

scanning confocal microscope (ZEISS, LSM880) and ZEN software (version 2012).

### Preparation of primary mouse hepatocytes

Primary mouse hepatocytes were isolated from three wild-type (WT) 8- to 12-week-old male C57BL/6J mice (Gempharmatech, Jiangsu, China) using collagenase perfusion and maintained in a collagen sandwich culture[36]. The primary mouse hepatocytes were cultured in 5% FBS-Williams' Medium E (GIBCO, Cat#12551) overnight, and were collected for total RNA extraction. The protocol of mouse-related experiments

was approved by the Institutional Animal Care and Use Committee of the First Affiliated Hospital of Third Military Medical University (AMUWEC20211648).

### RNA extraction, and quantitative RT-PCR

Total RNA was extracted from the isolated cholangiocytes using Trizol reagent (Takara Bio, Cat#9109) and reversely transcribed into cDNA for RT-qPCR analysis using the primers provided in Supplementary Table 15. Glyceraldehyde-3-phosphate dehydrogenase (*GAPDH*) was used as an internal control. The data were analyzed by $2^{-\Delta\Delta Ct}$.

**Fig. 5 | Interactions between DUOX2+ACE2+ small cholangiocytes and immune cells or cholangiocytes in the cluster (1) and (2). a** Cell-cell interaction networks predicted the potential interaction magnitude between cholangiocytes in the cluster (3) (named DUOX2+ACE2+ small cholangiocytes) and different types of immune cells in control and PBC livers ($n = 4$ and $n = 5$, respectively). Line thicknesses denote the numbers of ligand-receptor pairs. MP mononuclear phagocytes, NK cells natural killer cells, DCs dendritic cells. **b** Heatmap displayed the total numbers of ligand-receptor pairs between DUOX2+ACE2+ small cholangiocytes and different types of immune cells. **c** Dot plot exhibited the significant ligand-receptor pairs involved in the interactions between DUOX2+ACE2+ small cholangiocytes and different types of immune cells. Ligand and cognate receptors are shown in the *y* axis; cell populations that express ligand and receptor are shown in the *x* axis. Circle sizes denote *p* values; colors (red, high; blue, low) denote average ligand and receptor expression levels in interacting subpopulations. Statistics: two-tailed permutation test without adjustment, 95% confidence interval. **d** Cell-cell interaction networks predicted the potential interaction magnitude between DUOX2+ACE2+ small cholangiocytes and cholangiocytes in the cluster (1) or (2) as in **a**. **e** Heatmap displayed the total numbers of ligand-receptor pairs between DUOX2+ACE2+ small cholangiocytes and cholangiocytes in the cluster (1) or (2). **f** Dot plot exhibited the significant ligand-receptor pairs involved in the interactions between DUOX2+ACE2+ small cholangiocytes and cholangiocytes in the cluster (1) or (2) as in **c**. Statistics: two-tailed permutation test without adjustment, 95% confidence interval.

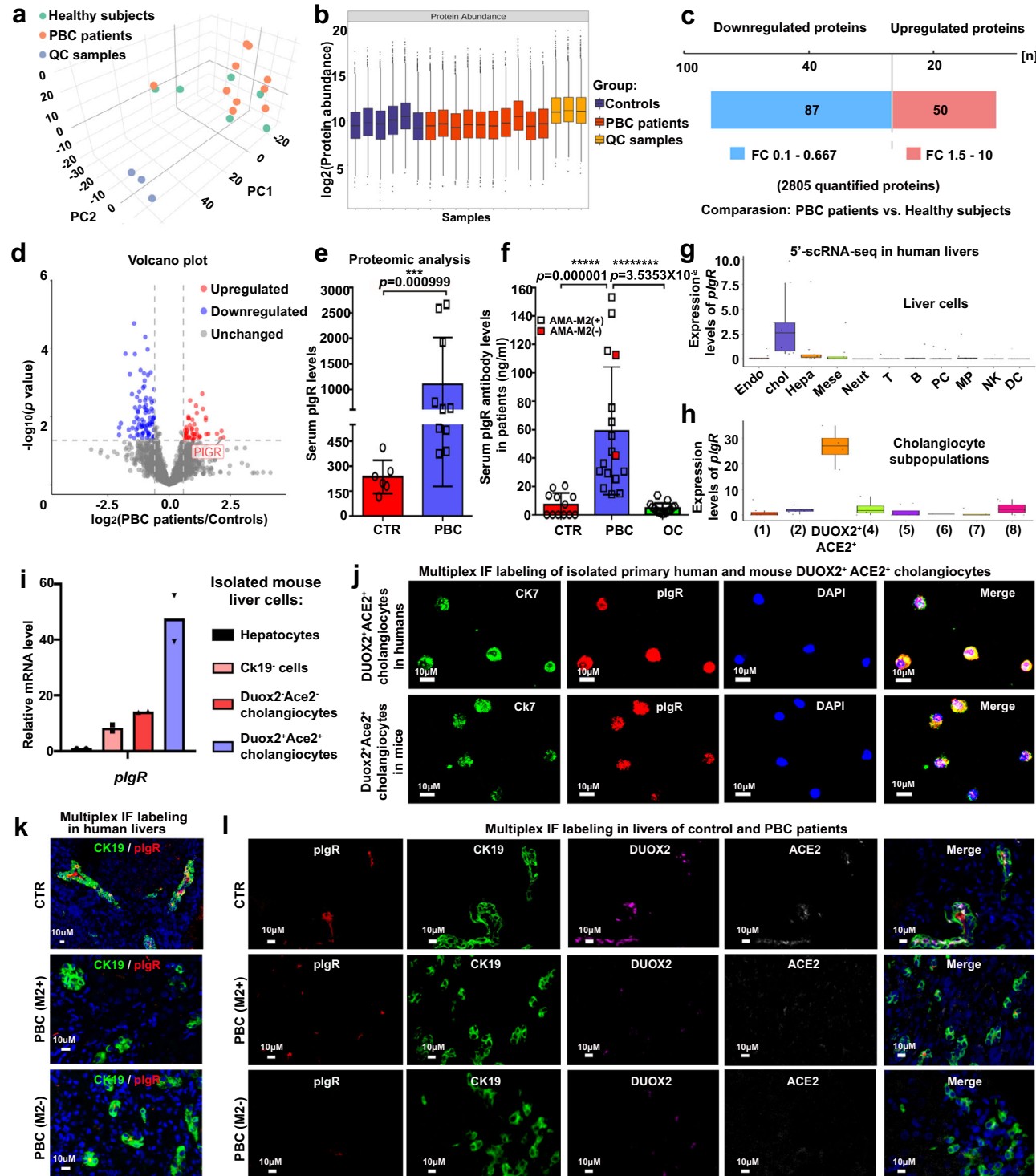

**Fig. 6 | The pIgR is mainly expressed in DUOX2$^+$ACE2$^+$ small cholangiocytes, while the levels of serum pIgR and its antibodies were significantly elevated in PBC patients. a** Score scatter plot corresponding to a principal components analysis (PCA) of the serum proteomics from six healthy subjects (CTR) and 10 primary biliary cholangitis (PBC) patients. QC quality control. **b** Quality control for CTR and PBC samples. The center line shows the median, the box limits represent the upper and the lower quartiles, and the whiskers extend to the largest and smallest values. CTR: $n = 6$ serum samples from CTR patients; PBC: $n = 10$ serum samples from PBC patients. **c** The Changes in serum proteomics in PBC patients and CTR subjects. Significantly changed proteins were identified with a fold-change (FC) greater than 1.5 and $p < 0.05$, statistically analyzed by two-tailed independent-sample Student's $t$ test. **d** Volcano plot of the changes in protein abundance between CTR and PBC patients. Statistics: two-tailed independent-sample Student's $t$ test, 95% confidence interval. **e** The levels of serum pIgR protein obtained from proteomic analysis. CTR: $n = 6$ serum samples from CTR patients, PBC: $n = 10$ serum samples from PBC patients; Plotted: mean ± SEM; Statistics: two-tailed Mann–Whitney $U$ test, 95% confidence interval; ***$p < 0.001$. **f** The levels of serum anti-pIgR autoantibodies in CTR, PBC, and obstructive cholestasis (OC) patients. CTR: $n = 12$ serum samples from CTR patients, PBC: $n = 17$ serum samples from PBC patients, OC: $n = 15$ serum samples from OC patients; Plotted: mean ± SEM; Statistics: two-tailed Mann–

Whitney $U$ test, 95% confidence interval; *****$p < 0.00001$, ********$p < 0.00000001$. **g**, **h** The pIgR was mainly expressed in cholangiocytes (**g**) and highly expressed in DUOX2$^+$ACE2$^+$ small cholangiocytes (**h**) in human liver from the analysis of 5′-scRNA-seq data. The center line shows the median, the box limits represent the upper and the lower quartiles, and the whiskers extend to the largest and smallest values. $n = 9$ human liver samples. Endo endothelial cells, Chol cholangiocytes, Hepa hepatocytes, Neut neutrophils, PC plasma cells, MP mononuclear phagocytes, NK natural killer cells, DC dendritic cells. **i** The relative levels of pIgR mRNA transcripts in Duox2$^+$Ace2$^+$ or Duox2$^-$Ace2$^-$ cholangiocytes, Ck19$^-$ cells and hepatocytes from normal mouse livers. $n = 2$ independent experiments. **j** Representative multiplex immunofluorescence (IF) photomicrographs of CK7 (green), pIgR (red) and DAPI (blue) in primary human DUOX2$^+$ACE2$^+$ small cholangiocytes and mouse Duox2$^+$Ace2$^+$ small cholangiocytes. Scale bars: 10 μm. $n = 3$ independent experiments. **k** Representative multiplex IF photomicrographs of CK19 (green) and pIgR (red) in liver sections of CTR and PBC patients with positive AMA-M2 or negative AMA-M2. Scale bars: 10 μm. $n = 3$ independent experiments. **l** Representative multiplex IF photomicrographs of pIgR (red), CK19 (green), DUOX2 (purple), ACE2 (gray white), and DAPI (blue) in liver sections of CTR and PBC patients with positive AMA-M2 or negative AMA-M2. Scale bars: 10 μm. $n = 3$ independent experiments. Source data are provided as a Source Data file.

## Liver histology

Liver sections were routine-stained with HE, Masson and Sirius Red. Human liver histological assessment was performed with the Nakanuma stage and the Ludwing stage[24,25].

## Immunohistochemistry analysis

Immunohistochemistry (IHC) analysis of CK7 expression in liver tissue sections was performed. The primary antibodies and their dilutions are presented in Supplementary Table 14.

## Proteomic analysis of serum samples in PBC patients

First, proteins from serum samples of six volunteers and 10 PBC patients were fractionated for data-dependent acquisition (DDA) and data-independent acquisition (DIA) analysis. Equal aliquots of these samples were pooled into one sample for generating a DDA library. The high and low abundant proteins in the pooled sample were separated and collected using Human 14 Multiple Affinity Removal System Column following the manufacturer's protocol (Agilent Technologies, Cat#5188-6557). The eluted proteins were filtrated using a 5 kDa ultrafiltration tube (Sartorius, Cat#VS04T12). The individual serum samples and the high and low abundant protein samples obtained from the pooled sample were solved in SDT buffer (4% SDS, 100 mM DTT, 150 mM Tris-HCl pH 8.0), boiled for 15 min and centrifuged at $14,000 \times g$ for 20 min. The supernatants were aliquoted and stored at −80 °C.

Second, the soluble proteins of individual serum samples and of the high and low abundant protein samples obtained from the pooled sample were digested, according to the filter-aided sample preparation (FASP) protocol[37]. Briefly, 200 μg proteins in UA buffer (8 M Urea, 150 mM Tris-HCl pH 8.0) were filtered through 10-Kd Microcon centrifugal filter unit to remove DTT and other low-molecular-weight components. After being washed with 100 μl of UA buffer for three times and with 100 μl of 25 mM $NH_4HCO_3$ buffer for twice, the proteins were digested with 4 μg trypsin (Promega, Cat#317107) in 40 μl of 25 mM $NH_4HCO_3$ buffer at 37 °C overnight, and the resulting peptides were collected, followed by filtration. The peptides were desalted on C18 cartridges (Empore™ SPE Cartridges C18 (standard density), bed I.D. 7 mm, volume 3 ml, Sigma, Cat#66872-U), concentrated by vacuum centrifugation and reconstituted in 40 μl of 0.1% (v/v) formic acid. The peptide contents were estimated by a UV light spectrometry at 280 nm using an extinction coefficient of 1.1 of 0.1% (g/l) solution. The digested peptides from low abundant protein sample were separated into 10 fractions using the High pH Reversed-Phase Peptide Fractionation Kit (Thermo Scientific™ Pierce™, Cat#84868). The peptides in each fraction were concentrated by vacuum centrifugation and

reconstituted in 15 μl of 0.1% (v/v) formic acid. The collected peptides were desalted on C18 cartridges and reconstituted in 40 μl of 0.1% (v/v) formic acid. The peptides were mixed with standard peptides in iRT-Kits (Biognosys, Cat#Ki-3002-2) at a ratio of 1:3 for MS analysis.

Third, mass spectrometry (MS) assay was performed. For DDA-MS analysis, all digested fractionated low and high abundant protein samples were analyzed by TIMSTOF mass spectrometer (Bruker Daltonics, Bremen, Germany) via an Evosep One system liquid chromatography (Evosep, Denmark). The peptides were loaded onto a reverse phase trap column (Thermo Scientific Acclaim PepMap100, 100 μm*2 cm, nanoViper C18) connected to the C18-reversed phase analytical column (Thermo Scientific Easy Column, 10 cm long, 75 μm inner diameter, 3 μm resin) in buffer A (0.1% Formic acid) and separated with a linear gradient of buffer B (84% acetonitrile and 0.1% Formic acid) at a flow rate of 300 nl/min controlled by IntelliFlow technology. The mass spectrometer was operated in the DDA mode for the ion mobility-enhanced spectral library generation. We set the accumulation and ramp time at 100 ms each and recorded mass spectra in the range from $m/z$ 100 to 1700 in positive electrospray mode, dynamic exclusion of 24.0 s. Ion source Voltage was set at 1500 V, temperature at 180 °C, dry gas at 3 L/min. The ion mobility was scanned from 0.6 to 1.6 Vs/cm$^2$, subjected to eight cycles of PASEF MS/MS using eight parallel accumulation-serial fragmentation (PASEF) MS/MS scans. For DIA-MS analysis, the digested individual serum protein samples were analyzed by the same mass spectrometer and liquid chromatography as used in the DDA-MS analysis in the DIA mode. The nanoLC conditions for the DIA-MS analysis were also the same with those for the DDA-MS analysis described above. The mass spectrometer collected ion mobility MS spectra over a mass range of $m/z$ 100–1700, with eight windows for single 100 ms TIMS scans. The collision energy during PASEF scanning was ramped linearly from 20 eV at 1/K0 = 0.6 Vs/cm$^2$ to 59 eV at 1/K0 = 1.6 Vs/cm$^2$.

Fourth, the obtained mass spectrometry data were analyzed. The DDA library data were firstly screened against the FASTA sequence database using the Spectronaut (version 14.4.200727.47784, https://www.biognosys.com/), including Uniprot_human database and iRT peptides sequence (Biognosys|iRT-Kits|) with the parameters of trypsin, max missed cleavages at 2, fixed modification of carbamidomethyl(C), dynamic modification of oxidation(M) and acetyl (Protein N-term). All reported peptides were determined by a false discovery rate (FDR = N (decoy) * 2 / (N (decoy) + N (target))) ≤ 1% with 99% confidence for protein identification. The original raw files and DDA searching results were exported into Spectronaut (version 14.4.200727.47784) to construct a spectral library. The DIA data in the spectral library were analyzed using the Spectronaut (version

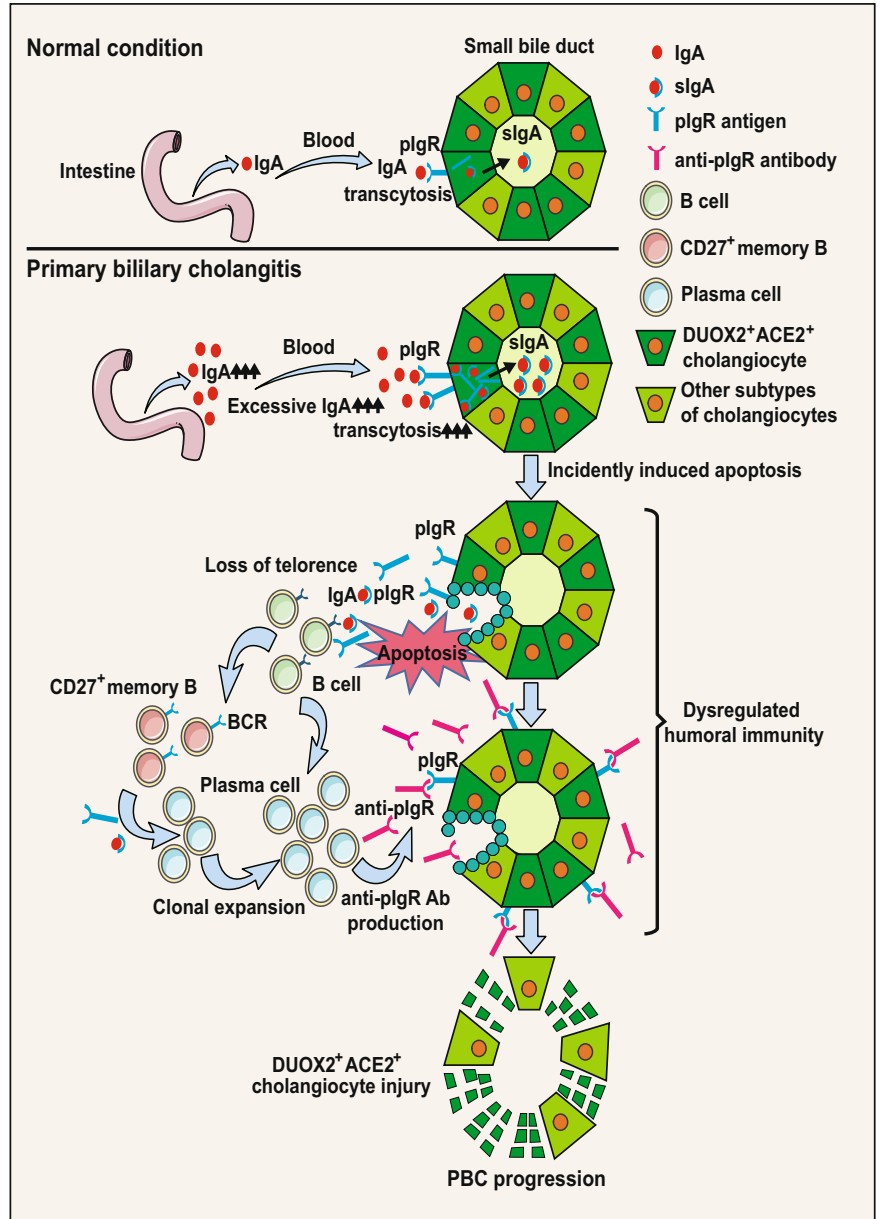

**Fig. 7 | A proposed immunological mechanism underlying the pathogenesis of PBC.** Initially, excessive IgA undergo transcytosis through the pIgR into DUOX2⁺-ACE2⁺ small cholangiocytes and incidentally induce cholangiocyte apoptosis. Subsequently, the apoptotic DUOX2⁺ACE2⁺ small cholangiocytes cause the leakage of pIgR autoantigen, leading to the loss of immune tolerance by inducing pIgR-specific humoral and T cell immunity, and recruiting CD27⁺ memory B and plasma cell infiltration in the hepatic portal tracks. The induced pIgR-specific auto-antibodies specifically damage liver DUOX2⁺ACE2⁺ small cholangiocytes in the small bile ducts to impair bile secretion, leading to PBC development. PBC primary biliary cholangitis, IgA immunoglobulin A, sIgA secretory immunoglobulin A, BCR B-cell receptor.

14.4.200727.47784) with the parameters of dynamic iRT for retention time prediction, enable interference on MS2 level correction, and enable cross-run normalization. All results were filtered through a Q value cutoff 0.01 (equivalent to FDR < 1%). All proteomic analyses were technically helped by Shanghai Applied Protein Technology.

Finally, the serum proteomic data were used in the following bioinformatic analyses. Cluster 3.0 (version 3.0, http://bonsai.hgc.jp/~mdehoon/software/cluster/software.htm) and Java Treeview software (version 3.0, http://jtreeview.sourceforge.net) were used for hierarchical clustering analysis. The hierarchical clustering was further analyzed by euclidean distance algorithm for measurement of similarity and average linkage clustering algorithm (using the centroids of the observations) for clustering. The difference in protein contents between groups was presented as a heat-map with a dendrogram.

Protein sequences were identified for protein domain signatures in the InterPro member database Pfam using the InterProScan software (version 5.25-64.0, http://www.ebi.ac.uk/interpro/download/InterProScan/). The interested proteins were blasted against the online KEGG database (http://geneontology.org/) to reveal their KEGG orthology identifications and were subsequently mapped to pathways in the KEGG. The potential functional categories and pathways of an interested protein were analyzed using the whole quantified proteins as background dataset and determined by the Fisher' exact test and Benjamini-Hochberg correction with a $p$-value of 0.05.

### ELISA
The levels of serum anti-pIgR antibodies in 12 control subjects, 17 PBC and 15 OC patients were determined by ELISA using an anti-pIgR

antibody ELISA Kit (AMOY LunChangShuo Biological Technology, Cat#100712), according to the manufacturer's instructions.

## Statistical analysis

All experiment data are expressed as mean ± standard error of mean (SEM). Statistical analysis was carried out with SPSS software (version 23) and graphing was performed using GraphPad Prism software (version 8). Two-tailed independent-sample Student's *t*-test or Mann–Whitney *U* test was used to determine the statistical significance of difference between two groups according to the results of the Shapiro-Wilk normality tests. A *p* value of <0.05 was considered statistically significant.

## Reporting summary

Further information on research design is available in the Nature Portfolio Reporting Summary linked to this article.

## Data availability

The raw data and their processed data of scRNA-seq, BCR and TCR sequencing, and spatial transcriptomics generated in this study have been deposited in the Genome Sequence Archive (GSA) under accession code HRA002347 and Open Archive for Miscellaneous (OMIX) under accession code OMIX001122, respectively. Proteomics data have also been deposited in the OMIX under accession code OMIX001127. All above data presented in the PRJCA009122 project of the National Genomics Data Center, China National Center for Bioinformation. Source data are provided with this paper.

## Code availability

The code used in this study can be found at: https://github.com/bobxjt/PBC.

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

## Acknowledgements

This work was financially supported by the grants from the National Natural Science Foundation of China (NSFC) (81922012 to J.C., 31971086 to Q.P.), the Outstanding Youth Science Foundation of Chongqing (cstc2021jcyj-jqX0005 to J.C.), the Project of Chongqing Universities Innovation/Outstanding Medical Research Group (2021cqspt01 and 4246ZO1 to J.C.), and Natural Science Foundation of Southwest Hospital and Third Military Medical University (2017YQRC-01 and XZ-2019-505-001 to J.C.). We thank the patients for their participation and cooperation.

## Author contributions

J.C., S.P., X.Liu, C.T., and X.H. designed the experiments; X.Li, Y.L., J.X., H.W., Y.G., X.M., and P.S. performed the experiments; X.Li, Y.L., J.X., H.W., Y.G., X.M., P.S., Y.H., X.Z., N.Z., and M.Z. analyzed the data; Y.H., J.D., Y.T., M.L., L.L., Y.P., X.L., Q.P., Q.X., Q.L., J.L., Y.L., Z.C., and Y.H. contributed to critical reagents/materials/analysis tools; J.C., S.P., X.Liu, C.T., X.H., and X.Li wrote and revised the manuscript. D.A., S.C., and J.B. contributed to the critical revision of the manuscript. All authors commented on the manuscript.

## Competing interests

The authors declare no competing interests.

## Additional information

[1]Department of Gastroenterology, Institute of Digestive Diseases of PLA, Cholestatic Liver Diseases Center, and Center for Metabolic Associated Fatty Liver Disease, the First Affiliated Hospital (Southwest Hospital) to Third Military Medical University (Army Medical University), Chongqing 400038, PR China. [2]Department of Hematology, the Third Affiliated Hospital (Daping Hospital), Third Military Medical University (Army Medical University), Chongqing 400042, PR China. [3]Department of Gastroenterology, Xiangya Hospital, Central South University, Changsha 410008, PR China. [4]Department of Hepatology and Infectious Diseases, Xiangya Hospital, Central South University, Changsha 410008, PR China. [5]Department of Endocrinology, Xiangya Hospital, Central South University, Changsha 410008, PR China. [6]MAFLD Research Center, Department of Hepatology, the First Affiliated Hospital of Wenzhou Medical University, Wenzhou 325035, PR China. [7]Institute of Hepatobiliary Surgery, the First Affiliated Hospital (Southwest Hospital), Third Military Medical University (Army Medical University), Chongqing 400038, PR China. [8]Institute of Hepatobiliary Surgery, the Second Affiliated Hospital (Xinqiao Hospital), Third Military Medical University (Army Medical University), Chongqing 400038, PR China. [9]Department of Hematology, the First Affiliated Hospital (Southwest Hospital), Third Military Medical University (Army Medical University), Chongqing 400038, PR China. [10]Department of Internal Medicine and Liver Center, Section of Digestive Diseases, Yale University School of Medicine, 333 Cedar Street, New Haven, CT 06520, USA. [11]Center of Minimally Invasive Intervention, the First Affiliated Hospital (Southwest Hospital), Third Military Medical University (Army Medical University), Chongqing 400038, PR China. [12]Institute of Medical Science Research, Xiangya Hospital, Central South University, Changsha 410008, PR China. [13]National Clinical Research Center for Geriatric Disorders, Xiangya Hospital, Central South University, Changsha 410008, PR China. [14]These authors contributed equally: Xi Li, Yan Li, Jintao Xiao, Huiwen Wang, Yan Guo, Xiuru Mao, Pan Shi. ✉e-mail: hxuequan@163.com; tangcane@csu.edu.cn; liuxw@csu.edu.cn; sfp1988@csu.edu.cn; jin.chai@cldcsw.org

