## [Peer Review File · Nature Communications]

Reviewers' Comments:

Reviewer #1:

Remarks to the Author:

This paper has the merit to be the first one to provide a human atlas in primary biliary cholangitis using RNA single cell methods and spatial transcriptomics. The results provided are interesting and suggest the new technologies might be useful to get a deeper insight in disease biology. However, scRNA seq experiments require careful study design and rigorous optimization and the results carefully interpreted and validated. I have multiple concerns with the methodological approach presented and with the results presentation.

1. The first issue is with the sample size: 5 human PBC liver remains a rather small population. Moreover, in the paper is not clarified the reason why these patients were biopsied, as all them were AMA positive and had an increased ALP. This can generate a considerable selection bias, especially in view of the small sample size.

2. The number of liver cells studied is relatively low. A total of 40,272 cells covers 30 cell populations, most of the cells should be hepatocytes and lymphoid infiltrates. In Figure 1H, the cholangiocytes in PBC only counted for about 1% of liver cells. It's about 400 cholangiocytes from PBC. Given there were 8 populations distributed in cholangiocytes (Figure 2C), only less than 10% of cholangiocytes in control subjects were in cluster cholangiocytes (3) (Figure 3D) the target population in the paper. That means only 40 cells in cholangiocytes cluster 3 in control subjects. Please note that only cluster 3 was positive for DUOX2 and ACE2 (Figure 2F).

The number doesn't make much sense if only 10% of cholangiocytes are the pathogenic targets of PBC.

However, the authors also demonstrated that about 55% of CK19 positive cells are DUOX2ARE positive cells (Figure 2G) . This number does not match the RNA seq data.

It would make sense, to be more selective and enrich (sorting?) for the cholangiocytes population, rather than analyze the whole tissue.

3. The present study could not exclude there might be other target cholangiocytes rather than this population. This population did not do much with immune response (Figure 4F).

4. I also question about the "pIgR exclusively expressed on DUOX2ARE positive small cholangiocytes, and the levels of serum anti-pIgR antibodies in PBC are significantly elevated". In Figure 6E, only 3 out of 10 PBC patients had increased serum levels of pIgR (IN RED).

Other epithelial cells express pIgR, DUOX2 and ACE2 eg. intestinal epithelial cells, but PBC does not target on gut.

5. The RNA seq data obtained from liver biopsy with UDCA untreated PBC patient, but I couldn't find the disease stage of these PBC.

6. I don't understand why there were IGKV, IGHV, IGLV genes detected in cholangiocytes (Figure 2F). Could it be the contamination of cholangiocytes? Why did the authors choose 2-OA model? What happens in mouse if they only induce DUOX2ARE positive cells damage? Any pIgR autoantibodies can be detected in mouse?

7. It has been previous chosen that enzymatic digestion affects the surface markers of immune cells (Autengruber A, Gereke M, Hansen G, Hennig C, Bruder D. Impact of enzymatic tissue disintegration on the level of surface molecule expression and immune cell function. Eur J Microbiol Immunol (Bp). 2012;2(2):112-120. doi:10.1556/EuJMI.2.2012.2.3). As such the results on immune system needs to be further validated with different isolation methods (including the T and B cell repertoires).

Reviewer #2:

Remarks to the Author:

The manuscript entitled "Unique DUOX2+ACE2+ small cholangiocytes are pathogenic targets of primary biliary cholangitis" described data showing that primary biliary cholangitis (PBC) affects mainly a subpopulation of cholangiocytes. For that, the authors have performed detailed single cell RNA-Seq analyses on patient biopsies and have identified missing population of cholangiocytes in diseased tissues. These cholangiocytes express ACE2 and DUOX2 and have been described previously as small cholangiocytes. They then showed that deletion of Duox2+Ace2+ small cholangiocytes induces the progression of PBC and cholestasis in mice. To further investigate the immunological aspects involved in the disease. They also used spatial transcriptomic on patient biopsies and uncovered that CD27+ memory B and plasma cells can accumulate in specific regions of the liver especially the portal area. Finally, they showed that DUOX2+ACE2+ small cholangiocytes specifically express hepatic pIgR is exclusively expressed and this expression could be associated with the decrease of the cells in PBC.

The manuscript presents interesting data supported by a broad number of methods. However, the manuscript will also benefit from some tidy up, image of better quality and some rewriting. Also, it would be helpful if they add a few experiments/information to support key data.

General comments:

More information is needed concerning the liver control patients used in their analyses. It is likely that these samples were obtained from patients with liver diseases/cancer which could interfere with the current interpretation.

More broadly, here presentation needs to be improved, for example image quality is often low, scale bars are missing on many images and in some cases they are essential for to support their claim on cell diameter. Generally, the manuscript needs to be better organised especially massive the data in the supplementary which are hardly referred to. Data should only be included if referred to in the text and they should be more specific on how they refer to data (at one point they reference 10 sup tables at once when many are not related to the text).

The interpretation on difference in subpopulation between diseases and control samples is misleading. Single cell dissociation efficacy will vary in function of disease status (i.e cholestatic, fibrosis, etc...) and this will strongly influence the cells collected. Some cells are extremely sensitive to dissociation especially hepatocytes and cholangiocytes. Thus, cell death is always extremely high for these cell types (explaining that natural ration of liver cells is not respected in those analyses. Hepatocytes should represent 80% of the cells). Similarly, the author should better explained how they can use spatial transcriptomic presented in Fig.5 to quantify number of cells. Indeed, such method has limited resolution (definitely not at the single cell). In addition, it is not clear if these analyses correlate the single cell RNA-Seq.

What is more intriguing is the apparent absence of disease signature in these analyses. It could be expected that PBC could have an impact on the gene expression profile of liver cells?

Figure 1 show that cholangiocytes express hepatocytes markers (Alb, TTR, Apoc1). This is a common issue encountered with single cell analyses of liver cells. Indeed, cell dissociation create a massive cell death in hepatocytes creating a soup of mRNA included in the drop process. Consequently, transcripts of genes highly expressed in hepatocytes are often detected in other cell type. The authors need to take this aspect in consideration especially when looking at specific markers in cholangiocytes.

Small cholangiocytes are defined as the cells located in the canal of hearing the extreme end of small bile duct. Thus, it should be a relatively rare population and not represent 55% of the cholangiocytes. The authors need to provide the negative controls used to define the gate of their FACs analyses in Figure 2G. They should also show the ACE2 analyses. This protein is often difficult to visualise by FACs/IF. It would be good to include more staining on sections of ACE2+ DUOX2 cells and examples of measurement to see these in the tissue as oppose to just as isolated

cells. A quantification of diameter in on sections ACE2+ DUOX2+ could be supportive data. This would fit well with Figure 3E and F (see below).

Small cholangiocytes have been broadly characterised previously. It would be useful if the authors could compare their small cholangiocytes using previously identified markers. They did perform QPCR on ssct and CFTR (Figure 3D) which are supposed to be differentially expressed in small vs large cholangiocytes. However, these genes seem to express at background level in both cell type. They should show all these markers using their single cell RNA-Seq data. Did they identify known small cholangiocytes markers in their DEG analyses?

Figure 3I are impossible to analyses. They are just too small and their resolution not good enough. It would be incredibly useful to show a bile duct with containing both KRT19/DUOX2+/ACE2+ and KRT19/DUOX2-/ACE2- cells in control. Indeed, all the KRT19+ cholangiocytes seems to express DUOX2 and ACE2.

The data on pIgR are really interesting but lack functional validation. It would be interesting to define if patient immune cells react specifically against cholangiocytes expressing this antigen.

Some specific points:

Figure 1H - It should be denoted which comparisons are significant and the full p value provided in the legend

Text – line 70 - In text they refer to many supplementary tables - this a huge amount of info much not discussed at this stage. The tables should only be referenced when appropriate to aid the reader in following flow of the paper

Text – “Interestingly, there was a unique cholangiocyte cluster (3) in control livers, but a few were detectable in PBC livers (Fig.2C-D).” This need to be reworded - be more specific and clear.

Figure 2G - FACS is confusing way it is presented. Lower panel is all Ck19+ as it is a daughter gate of the upper panel. Should not be shown as a quadrant, a single line should partition the ACE+ and ACE2-.

Figure 3E and F - are missing scale bars. More images should be shown with examples of small, medium and large ducts shown if possible, so the differences can be appreciated. Scale bars are necessary for this interpretation.

Imaging quality is low and appears out of focus, clearer images should be taken.

Figure 3I - Also missing scale bars and images could be improved

Figure 4B - Single fluorescent channels should be shown to help with interpretation

Figure 5C - spatial transcriptomics - wouldn't they expect that the proportion of cell samples on the sections which are hepatocytes to be higher - at least in the control?

Figure 5F - no scale bars. Single channels should be shown at least for the triple stain to help with interpretation

Figure 5G - it is unclear to me what this analysis is showing and what N represents. This should be explained In rebuttal and in the methods.

Reviewer #3:

Remarks to the Author:

The manuscript “NCOMMS-21-21721-T” entitled "Unique DUOX2+ACE2+ small cholangiocytes are

pathogenic targets of primary biliary cholangitis" presents a study of the chronic autoimmune liver disease called primary biliary cholangitis (PBC). Among the multiple analytical techniques utilized, the authors performed proteomics analyses by data-independent acquisition (DIA) mass spectrometry (MS). Overall, the proteomics experiments and results are well described. The following points are recommended to be addressed to improve the manuscript:

1. Proteomics is not explicitly mentioned in the abstract.
2. Please clarify how the individual samples were prepared for the DIA-MS analysis, since the described digestion and fractionation is referring to the pooled samples used to build the DDA library.
3. The nanoLC conditions used for the DIA-MS analysis should be included.
4. The instrument used for DIA-MS analyses can be inferred from the description but it should be clearly specified including the vendor, in the same way the instrument used to build the DDA-MS library was described.

Reviewer #4:

Remarks to the Author:

This manuscript identifies a sub-population of human cholangiocytes that specifically express ACE2, DUOX2 and PIGR and is enriched in bile secretion processes. The authors claim that these cholangiocyte subtypes are the target of autoimmune attack in PBC, giving rise to the associated portal inflammation. They further claim that the mechanism involves the death of these cells through excessive transfer of Iga, the consequent release of PIGR proteins and the stimulation of the production of auto-antibodies against PIGR. They perform proteomic measurements in blood to validate an increase in PIGR and in auto-antibodies against PIGR, supporting the model.

The manuscript is very interesting and proposes a novel mechanism that may underlie PBC. However, some of the key claims in the paper are not sufficiently supported by the data and should either be substantiated with additional experiments and analyses or removed. The following points must be addressed:

1) The authors claim that the ACE2+DUOX2+ cholangiocytes constitute 55% of cholangiocytes in control human livers. This finding is not supported by the data. From our analysis of the single cell 3'-UTR single cell data the cells only appear in one of the two control patients, with very low proportions in the second control patient. This is also consistent with previous human liver single cell atlases, in which these cells appear in extremely low proportions, as can be readily verified in the following browsers: <http://human-liver-cell-atlas.ie-freiburg.mpg.de/>, <https://shiny.igmm.ed.ac.uk/livercellatlas/>, <https://itzkovitzwebapps.weizmann.ac.il/webapps/home/session.html?app=HumanLiverBrowser>. The authors should present a table with the proportions of the ACE2+DUOX2+ cells in each patient as well as UMAPS colored by patient. They should also at least hypothesize why the abundances of these cells is so variable and why have they not been observed in previous human liver atlas studies (different cohorts? Different cell extraction protocols)?

2) A related note – the in-situ validations of these cells is not convincing. For example, Figure 3E seem to show 100% of cholangiocyte positive for the ACE2 and DUOX2 proteins. There are no scale bars in any images, the images are low resolution. Please include high magnification high resolution blow-ups. It is unclear if the positive cells are interspersed between the negative cells or rather cluster in distinct bile ducts, please clarify. Most importantly, little information is given on the antibody validation. The authors should perform RNA in-situ hybridizations for the relevant genes of interest, or conversely use well-validated antibodies. ACE2 and DUOX2 are expressed in intestinal villi, and so showing staining on such a tissue with established expression of the proteins is important for such validation. Figure 5F – again, in-situ quantification is problematic, no scale bars, not clear if CK19 has a high background or if we are seeing multiple bile ducts. Single molecule RNA in-situ hybridization (e.g. RNAScope) has substantially lower background and is therefore the gold standard.

3) Mouse experiments – the authors claim that ablation of Ace2+Duox2+ cholangiocytes promotes cholestasis. For this direction to be relevant to the establishment of the functional importance of these cells, the authors should first demonstrate the existence of these cells in mouse liver via one of many available single cell mouse liver atlases (e.g.

<https://www.sciencedirect.com/science/article/pii/S1097276519305830#app2>). From brief analysis I see a very small minority of mouse cholangiocytes expressing these markers. In

addition, the in-situ validation of the existence of the Duox2+ mouse cholangiocytes must be expanded (Figure 4B shows one example and quantification is unclear). A fundamental problem with the ablation experiment is that Duox2 and Ace2 are highly expressed in the intestinal epithelium. If the ablation yields decrease in intestinal epithelial cells, this would lead to systemic inflammation. If this piece of evidence is to be used, the authors must demonstrate that intestinal epithelial cells are not damaged and that the Duox2+ cholangiocytes are lost upon ablation. In my view, this direction is problematic and might need to be removed from the paper (the other findings of the existence of the human ACE2+DUOX2+ in even a small sub-population and its potential role in human PBC pathology is interesting enough in my view, considering that the other points are addressed).

4) Spatial transcriptomics – Analysis is problematic, hepatocytes seem to take up around 20% of the spot transcriptomes, this cannot be, it should approach 80%, please clarify or analyze differently.

5) Figure 5G analysis is unclear, please elaborate.

Point-by-Point Responses

Responses to the Reviewer 1:

This paper has the merit to be the first one to provide a human atlas in primary biliary cholangitis using RNA single cell methods and spatial transcriptomics. The results provided are interesting and suggest the new technologies might be useful to get a deeper insight in disease biology. However, scRNA seq experiments require careful study design and rigorous optimization and the results carefully interpreted and validated. I have multiple concerns with the methodological approach presented and with the results presentation.

Response: We greatly appreciate his/her positive comments and helpful suggestions for our manuscript, which allowed us to improve the quality of our manuscript.

1. *The first issue is with the sample size: 5 human PBC liver remains a rather small population.*

Response: Since the etiology of PBC remains elusive, we used scRNA-seq to identify potential targets. We chose 5 PBC samples that allowed us to perform statistical analysis on the obtained data. Similar small sample sizes (3~5 tissue samples) for scRNA-seq have been previously used in many high quantity studies [Ref.1-4].

References:

1. Ramachandran, P. et al. Resolving the fibrotic niche of human liver cirrhosis at single-cell level. *Nature* **575**, 512-518 (2019).
2. Corridoni, D. et al. Single-cell atlas of colonic CD8+ T cells in ulcerative colitis. *Nat. Med.* **26**, 1480-1490 (2020).
3. MacParland, S.A. et al. Single cell RNA sequencing of human liver reveals distinct intrahepatic macrophage populations. *Nat. Commun.* **9**, 4383 (2018).
4. Zhang, M. et al. Single-cell transcriptomic architecture and intercellular crosstalk of human intrahepatic cholangiocarcinoma. *J. Hepatol.* **73**, 1118-1130 (2020).

Moreover, in the paper is not clarified the reason why these patients were biopsied, as all them were AMA positive and had an increased ALP. This can generate a considerable selection bias, especially in view of the small sample size.

Response: We apologize for the confusion. The major reason for biopsy in these PBC patients was to exclude autoimmune hepatitis and other liver diseases. We have modified the text and clarified the reason in the Method section of the revised manuscript (page 17, lines 373-374).

2. The number of liver cells studied is relatively low. A total of 40,272 cells covers 30 cell populations, most of the cells should be hepatocytes and lymphoid infiltrates.

Response: We thank you for your critical comments and professional suggestions,

which allowed us to improve the manuscript. We would like to clarify that we obtained 70,050 liver cells, including 40,272 cells from 5 PBC biopsied liver samples and 29,778 cells from 4 control resected liver samples for 5'-scRNA-seq analysis. We believe these numbers are adequate for us to carry out the screening analysis. The numbers of isolated liver cells were higher in the current study than in some recently published studies. For example, McParland *et al* reported 8,444 liver cells from 5 normal liver tissues [Ref.1], and Zhang *et al* used 31,302 liver cells from 3 adjacent tissues and 5 intrahepatic cholangiocarcinoma for scRNA-seq analysis [Ref.2]. Next, these liver cells were divided into different cell subtypes, including hepatocytes (28.72%), lymphocytes (29.45%) and cholangiocytes (5.55%) in liver cells from human control livers, which were similar to a recent study (Supplementary Table 5) [Ref.1].

References:

1. McParland, S.A. et al. Single cell RNA sequencing of human liver reveals distinct intrahepatic macrophage populations. *Nat. Commun.* **9**, 4383 (2018).
2. Zhang, M. et al. Single-cell transcriptomic architecture and intercellular crosstalk of human intrahepatic cholangiocarcinoma. *J. Hepatol.* **73**, 1118-1130 (2020).

In Figure 1H, the cholangiocytes in PBC only counted for about 1% of liver cells. It's about 400 cholangiocytes from PBC. Given there were 8 populations distributed in cholangiocytes (Figure 2C), only less than 10% of cholangiocytes in control subjects were in cluster cholangiocytes (3) (Figure 3D) the target population in the paper. That

means only 40 cells in cholangiocytes cluster 3 in control subjects. Please note that only cluster 3 was positive for DUOX2 and ACE2 (Figure 2F). The number doesn't make much sense if only 10% of cholangiocytes are the pathogenic targets of PBC.

Response: After studied his/her critical comments, we have carefully examined our experimental data, and performed additional data analysis. In our 5'-scRNA-seq data, there were 1653 cholangiocytes accounting for 5.55% of liver cells from human control livers, similar to the data reported previously in human normal liver tissues [Ref. 1]. However, in the PBC livers, there were a total of 556 cholangiocytes, accounting for 1.38% of the total cell population (Supplementary Table 5). This makes sense because loss of cholangiocytes is a typical finding of PBC. Further analysis of cholangiocyte subpopulations revealed 308 cholangiocyte cluster (3) cells (named DUOX2⁺ACE2⁺ cholangiocytes), accounting for 18.63% of all cholangiocytes (1653 cholangiocytes) in human control livers, whereas there was few detectable DUOX2⁺ACE2⁺ cholangiocytes in PBC liver tissues (Supplementary Table 6). This data indicates that DUOX2⁺ACE2⁺ cholangiocytes are crucial for the pathogenesis of PBC. Most importantly, liver histological analysis revealed that a selective decrease in the number of DUOX2⁺ACE2⁺ small cholangiocytes in the liver of PBC patients was significantly associated with the severity of PBC (Fig. 3c-f), supporting the importance of DUOX2⁺ACE2⁺ cholangiocytes in the pathogenesis of PBC. In addition, we also observed decreased numbers of cholangiocyte clusters (1) and (2) in PBC patients compared to control patients (Supplementary Table 6). However, there were no

significant differences in the proportion of these cholangiocytes between the control and PBC groups (Fig. 2d). Furthermore, analysis of cell-cell interactions using CellPhoneDB in our 5'-scRNA-seq data predicted significant ligand-receptor interactions between DUOX2⁺ACE2⁺ small cholangiocytes and cholangiocyte cluster (1) or (2) cells (new Fig. 5d-f), suggesting that DUOX2⁺ACE2⁺ small cholangiocytes affect the survival of other subtypes of cholangiocytes during the pathogenic process of PBC. It is notable that PBC is an autoimmune disease and that early primed autoimmune responses can spread to other antigens intra- and inter-molecularly, driving the pathogenic process of an autoimmune disease, which may happen in PBC. However, whether and how cholangiocytes (1) and (2) subpopulations contribute to the pathogenesis of PBC needs to be investigated in future studies. We have added this new data in the revised manuscript.

References:

1. McParland, S.A. et al. Single cell RNA sequencing of human liver reveals distinct intrahepatic macrophage populations. *Nat. Commun.* **9**, 4383 (2018).

However, the authors also demonstrated that about 55% of CK19 positive cells are DUOX2ACE positive cells (Figure 2G). This number does not match the RNA seq data. It would make sense, to be more selective and enrich (sorting?) for the cholangiocytes population, rather than analyze the whole tissue.

Response: We greatly appreciate his/her critical comments and constructive suggestions. To address the concern, we have performed additional FACS experiments for three times by using corresponding fluorescein conjugated isotype control antibodies (Fig. 2g and Supplementary Table 14). As shown in Fig. 2g, FACS analysis demonstrated that DUOX2⁺ACE2⁺ cholangiocytes accounted for 18.7% of CK19⁺ cells in human liver samples and 22.2% of CK19⁺ cells in mouse liver samples. These experimental results were similar to the data analyzed previously from the 5'-scRNA-seq analysis in human liver samples (Fig. 2d and Supplementary Table 6). We have added this new experimental data and isotype control antibody information to the revised manuscript and supplementary documents.

3. The present study could not exclude there might be other target cholangiocytes rather than this population. This population did not do much with immune response (Figure 4F).

Response: We fully agree with the reviewer's comments. As shown in Fig.2d, cholangiocyte cluster (3) cells (named DUOX2⁺ACE2⁺ cholangiocytes) accounted for 18.63% of all cholangiocytes in human control livers, but were completely undetectable in PBC livers (Supplementary Table 7). Liver histologic analysis revealed that a dramatic decrease in the numbers of DUOX2⁺ACE2⁺ small cholangiocytes in PBC patients was significantly associated with the severity of PBC (Fig. 3c-f). These data support our conclusion that DUOX2⁺ACE2⁺ cholangiocytes are crucial for the

pathogenesis of PBC. Further analysis of cell-cell interactions by using CellPhoneDB in our 5'-scRNA-seq data predicted the significant ligand-receptor interactions between DUOX2⁺ACE2⁺ small cholangiocytes and immune cells (new Fig. 5a-c).

In addition, we also observed decreased numbers of cholangiocytes (1) and cholangiocytes (2) in PBC patients (Supplementary Table 6). However, their corresponding proportions in all cholangiocytes were not significantly changed in PBC patients compared to control patients (Fig. 2d). Data analysis indicates that there are significant cell-cell interactions between DUOX2⁺ACE2⁺ cholangiocytes (or immune cells) and cholangiocyte (1) or (2) cells (new Fig.5d-f and Supplementary Figure 14). These interactions may orchestrate the immune response and contribute to the development of PBC. We have added these new data in the revised manuscript .

4. I also question about the "pIgR exclusively expressed on DUOX2ACE positive small cholangiocytes, and the levels of serum anti-pIgR antibodies in PBC are significantly elevated". In Figure 6E, only 3 out of 10 PBC patients had increased serum levels of pIgR (IN RED).

Response: We apologize for the confusing interpretation of the figure due to the colour scale of the heatmap, and appreciate that the reviewer has pointed this out. Indeed, we found significant elevations of serum pIgR antigen in PBC patients ($n = 10$) compared to control patients ($n = 6$) by proteomic analysis (new Fig. 6e). The protein

abundance of pIgR in each patient is presented in the Supplementary Table 11. Furthermore, this elevation was also displayed by the proteomic heatmap (the original Fig.6e). To minimize any potential confusion and or redundance, we have removed the heatmap from the new Fig. 6.

In addition, we also found that the word “exclusively” is not accurate because pIgR was highly expressed in DUOX2⁺ACE2⁺ cholangiocytes while lowly expressed in other types of cholangiocytes (new Fig. 6h). We have changed the word “exclusively” to “highly” in the revised manuscript.

Other epithelial cells express pIgR, DUOX2 and ACE2 eg. intestinal epithelial cells, but PBC does not target on gut.

Response: Thank you for the insightful comment. We have also asked this question ourselves. We think it is most likely that the bile duct epithelium is facing a much more hostile environment than intestinal cells, since the bile contains millimolar concentrations of bile acids in bile lumen. Interestingly, a recent case report indicates that gut injury is seen not in all PBC patients but in some PBC patients, who also have inflammatory bowel disease (IBD) [Ref.1]. This would support the idea that autoantibody against pIgR plays a role in PBC and some patients with IBD, although more studies are needed. We have discussed this issue in the revised text.

References

1. Liberal, R. et al. Primary biliary cholangitis in patients with inflammatory bowel disease. *Clin. Res. Hepatol. Gastroenterol.* **44**, e5-e9 (2020).

5. *The RNA seq data obtained from liver biopsy with UDCA untreated PBC patient, but I couldn't find the disease stage of these PBC.*

Response: Thank you for pointing out this issue. Now, we have added the disease stage of PBC patients in the new Supplementary Table 2

6. *I don't understand why there were IGKV, IGHV, IGLV genes detected in cholangiocytes (Figure 2F). Could it be the contamination of cholangiocytes?*

Response: After performing additional analysis of our 5'-scRNA-seq data, we found that the expressions of IGKV, IGHV and IGLV were very low in cholangiocytes (Reply letter Fig. 1) as previously reported [Ref.1]. Moreover, further analysis of human liver scRNA-seq data in The Human Protein Atlas, a public database, also indicated that low levels of these IG gene transcripts in cholangiocytes (Reply letter Fig. 2). However, in order to minimize any confusion, we have removed these genes from the revised Fig. 2e, f.

Reply letter Fig. 1: The levels of cholangiocyte marker and IG-related gene transcripts in cholangiocytes in the 5'-scRNA-seq data

Reply letter Fig. 2: The levels of IGKV, IGHV and IGLV gene transcripts in cholangiocytes in human liver scRNA-seq data from The Human Protein Atlas.

References:

1. Shao, W. et al. Identification of Liver Epithelial Cell-derived Ig Expression in μ chain-deficient mice. *Sci. Rep.* **6**, 23669 (2016).

Why did the authors choose 2-OA model? What happens in mouse if they only induce DUOX2ARE positive cells damage? Any pIgR autoantibodies can be detected in mouse?

Response: Thanks for your critical comments and constructive suggestions. The 2-octanoic acid (2-OA) mouse model is characterized by lymphocytic infiltration and bile duct damage in liver portal area, accompanied by the elevated levels of serum AMA [Ref.1], similar to some characteristics in human PBC. Therefore, we used this mouse model to investigate the functional role of DUOX2⁺ACE2⁺ cholangiocytes. Indeed, we found a slight elevation of serum ALT and anti-pIgR autoantibodies in 2-OA mouse model (Reply letter Fig.3). However, according to the Reviewer #4's advice, we have removed these animal data from the revised manuscript due to insufficient data support.

Reply letter Fig. 3: The levels of alanine aminotransferase (ALT) **a** and anti-pIgR

antibodies (anti-pIgR) **b** in 2-OA mouse model without or with the deletion of DUOX2⁺ cholangiocytes.

References:

1. Katsumi, T. et al. Animal Models of Primary Biliary Cirrhosis. *Clin. Rev. Allergy Immunol.* **48**,142-153 (2015).

7. *It has been previous chosen that enzymatic digestion affects the surface markers of immune cells (Autengruber A, Gereke M, Hansen G, Hennig C, Bruder D. Impact of enzymatic tissue disintegration on the level of surface molecule expression and immune cell function. Eur J Microbiol Immunol (Bp). 2012;2(2):112-120. doi:10.1556/EuJMI.2.2012.2.3). As such the results on immune system needs to be further validated with different isolation methods (including the T and B cell repertoires).*

Response: After having read your comment, we realized this misinterpretation. In fact, we analyzed immune cells and immune repertoires, based on the transcription levels of immune cell related genes and the VDJ gene. We have modified the description of single cell preparation in the Method section.

Responses to the Reviewer 2:

The manuscript entitled “Unique DUOX2+ACE2+ small cholangiocytes are pathogenic targets of primary biliary cholangitis” described data showing that primary biliary cholangitis (PBC) affects mainly a subpopulation of cholangiocytes. For that, the authors have performed detailed single cell RNA-Seq analyses on patient biopsies and have identified missing population of cholangiocytes in diseased tissues. These cholangiocytes express ACE2 and DUOX2 and have been described previously as small cholangiocytes. They then showed that deletion of Duox2+Ace2+ small cholangiocytes induces the progression of PBC and cholestasis in mice. To further investigate the immunological aspects involved in the disease. They also used spatial transcriptomic on patient biopsies and uncovered that CD27+ memory B and plasma cells can accumulate in specific regions of the liver especially the portal area. Finally, they showed that DUOX2+ACE2+ small cholangiocytes specifically express hepatic pIgR is exclusively expressed and this expression could be associated with the decrease of the cells in PBC.

The manuscript presents interesting data supported by a broad number of methods. However, the manuscript will also benefit from some tidy up, image of better quality and some rewriting. Also, it would be helpful if they add a few experiments/information to support key data.

Response: We thank his/her positive and critical comments. We have performed additional experiments as you suggested.

General comments

1. *More information is needed concerning the liver control patients used in their analyses. It is likely that these samples were obtained from patients with liver diseases/cancer which could interfere with the current interpretation.*

Response: We appreciate his/her professional advice. We had debated regarding the control livers before the project started. To minimize/avoid any potential interference by other liver diseases as much as possible, we collected liver samples resected from patients with normal liver function tests due to hepatic hemangioma, hepatolithiasis, pancreatic or biliary tract tumour, etc. The detail information has been added in the Supplementary Tables 1, 8, 9, 12. Most importantly, in our 5' scRNA-seq data, 1653 cholangiocytes accounting for 5.55% of liver cells were identified in human control livers (Supplementary Table 5), similar to the previously reported data in normal human liver tissues [Ref. 1].

References:

1. McParland, S.A. et al. Single cell RNA sequencing of human liver reveals distinct intrahepatic macrophage populations. *Nat. Commun.* **9**, 4383 (2018).

2. *More broadly, here presentation needs to be improved, for example image quality is often low, scale bars are missing on many images and in some cases they are essential for to support their claim on cell diameter.*

Response: Thank you for pointing this out. Now, we have added scale bars to all images and tried our best to improve the quality of the images in the revised manuscript.

Generally, the manuscript needs to be better organized especially massive the data in the supplementary which are hardly referred to. Data should only be included if referred to in the text and they should be more specific on how they refer to data (at one point they reference 10 sup tables at once when many are not related to the text).

Response: We appreciate his/her advice. We have carefully re-organized the manuscript and modified the text to ensure that all data were correctly cited in the revised version.

3. The interpretation on difference in subpopulation between diseases and control samples is misleading. Single cell dissociation efficacy will vary in function of disease status (i.e cholestatis, fibrosis, etc...) and this will strongly influence the cells collected. Some cells are extremely sensitive to dissociation especially hepatocytes and cholangiocytes. Thus, cell death is always extremely high for these cell types (explaining that natural ration of liver cells is not respected in those analyses. Hepatocytes should represent 80% of the cells).

Response: We agree that dissociation affects scRNA-seq results. Bearing this in mind, to reduce variation of single cell dissociation before the scRNA-seq analysis, we

initially used control livers that were resected from patients with normal liver function tests and biopsied liver samples from PBC patients with the same histologic stages (Nakanuma stage II) (Supplementary Table 2). Secondly, the proportion of hepatocytes in liver cells remains relatively low as described previously [Refs.1,2] even when we have re-analyzed our scRNA-seq data. We think that this result may be attributed to: (1) Agreeing to the reviewer's comment, hepatocytes are more sensitive to the process of dissociation, resulting in cell death; and (2) The diameter of isolated hepatocytes with GEMs (form Gel Beads-in-emulsion) approximates the diameter of single cell microfluidic channels (10X Genomics experiment requirements: the cell diameter < 40µM), causing the loss of hepatocytes in the experimental process. In addition, hepatocytes represent ~80% of liver volume because their sizes are bigger than other cells in the liver [Ref.3]. However, their number consists of ~60% of liver cells [Ref.4]. To avoid the misleading interpretation for hepatocytes, we have removed the analyzed data related to hepatocytes from the scRNA-seq in the revised manuscript and supplementary documents.

References:

1. Sun, X.J. et al. Transcriptional switch of hepatocytes initiates macrophage recruitment and T cell suppression in endotoxemia. *J. Hepatol.* **S0168-8278**, 00136-2 (2022).
2. Aizarani, N. et al. A human liver cell atlas reveals heterogeneity and epithelial progenitors. *Nature* **572**,199-204 (2019).

3. Blouin, A., Bolender, R.P., Weibel, E.R. Distribution of organelles and membranes between hepatocytes and nonhepatocytes in the rat liver parenchyma. A stereological study. *J Cell Biol.* **72**, 441-455 (1977).
4. Kumar, S., Duan, Q., Wu, R., Harris, E.N., Su, Q. Pathophysiological communication between hepatocytes and non-parenchymal cells in liver injury from NAFLD to liver fibrosis. *Adv. Drug Deliv. Rev.* **176**, 113869 (2021).

Similarly, the author should better explain how they can use spatial transcriptomic presented in Fig.5 to quantify number of cells. Indeed, such method has limited resolution (definitely not at the single cell). In addition, it is not clear if these analyses correlate the single cell RNA-Seq.

Response: We thank his/her comments and appreciate her/his helpful advice. We combined the results of scRNA-seq to identify cell type in each spot of spatial transcriptomic using the method of deconvolution as described previously [Ref.1]. The detail for the procedures are described as follows: The expression profile of each spot in the spatial transcriptome is regarded as the expression profile of different cells after mixing; taking the expression profile of different cell types obtained in the scRNA-seq as a reference, the cell composition in each spot were inferred. The detailed implementation process of the method is as follows: Deconvolution was started with Seurat label transforming. Data with cell types annotated from scRNA-seq and spatial transcriptomic data did the re-normalization and PCA by Seurat using SCTransform

and RunPCA. Then, FindTransferAnchors was used to find anchors for integration with the two datasets. The underlying composition of cell types was predicted to deconvolute each of the spatial voxels by TransferData method. Through the steps above, a table containing the proportions of each cell types in spatial transcriptomic data was generated. This table was passed to stLearn to classify and visualize the cell type composition in each cluster. We have added the above description and relative references in the Methods section in the revised manuscript.

References:

1. Mantri, M. et al. Spatiotemporal single-cell RNA sequencing of developing chicken hearts identifies interplay between cellular differentiation and morphogenesis. *Nat. Commun.* **12**, 1771 (2021).

4. *What is more intriguing is the apparent absence of disease signature in these analyses. It could be expected that PBC could have an impact on the gene expression profile of liver cells?*

Response: We appreciate your professional comments. There were significant alterations in gene expression in each subpopulation of liver cells between control and PBC group (Supplementary Fig. 5). However, the associations between these altered genes and the progress of PBC needs to be investigated in our next project. Accordingly, these new data have been added in the revised manuscript.

5. *Figure 1 show that cholangiocytes express hepatocytes markers (Alb, TTR, Apoc1). This is a common issue encountered with single cell analyses of liver cells. Indeed, cell dissociation create a massive cell death in hepatocytes creating a soup of mRNA included in the drop process. Consequently, transcripts of genes highly expressed in hepatocytes are often detected in other cell type. The authors need to take this aspect in consideration especially when looking at specific markers in cholangiocytes.*

Response: Thank you for your comments and advice. We have excluded these hepatocyte genes when selected specific markers for cholangiocytes in data analysis in the revised manuscript (Fig. 2e, f).

6. *Small cholangiocytes are defined as the cells located in the canal of hearing the extreme end of small bile duct. Thus, it should be a relatively rare population and not represent 55% of the cholangiocytes. The authors need to provide the negative controls used to define the gate of their FACs analyses in Figure 2G. They should also show the ACE2 analyses. This protein is often difficult to visualise by FACs/IF.*

Response: Based on the reviewer's advice, we have performed additional FACS experiments for three times by using corresponding fluorescein conjugated isotype control antibodies (Fig. 2g and Supplementary Table 14). Our new FACS data indicated that DUOX2⁺ACE2⁺ cholangiocytes accounted for 18.7% of all cholangiocytes in human control livers and 22.2% of cholangiocytes in normal mouse

livers, respectively (Fig. 2g). According to the reviewer's suggestion, we have displayed the ACE2 analysis in the revised Fig. 2g. Moreover, we also performed RNAScope (the single molecule RNA in-situ hybridization) to confirm our multiplex IF results. As expected, RNAScope data supported our observation in the multiplex IF analysis (Fig. 2h). We have added these new data in the revised manuscript and supplementary documents.

It would be good to include more staining on sections of ACE2+ DUOX2+ cells and examples of measurement to see these in the tissue as oppose to just as isolated cells. A quantification of diameter in on sections ACE2+ DUOX2+ could be supportive data. This would fit well with Figure 3E and F (see below).

Response: Thank you for the constructive suggestions. We have now displayed more ACE2+DUOX2+ cholangiocytes in liver sections (new Fig. 3c, g) and cell slides (Fig. 2i). Moreover, we have also added scale bars to measure their diameters in liver sections (new Fig. 3c, g).

7. Small cholangiocytes have been broadly characterized previously. It would be useful if the authors could compare their small cholangiocytes using previously identified markers. They did perform QPCR on SCTR and CFTR (Figure 3D) which are supposed to be differentially expressed in small vs large cholangiocytes. However, these genes seem to express at background level in both cell type. They should show all these

markers using their single cell RNA-Seq data. Did they identify known small cholangiocytes markers in their DEG analyses?

Response: We have studied your thoughtful comments with care. As shown in Supplementary Fig.9, DEGs analysis displayed that DUOX2⁺ACE2⁺ cholangiocytes highly expressed the small cholangiocyte marker genes, including FOXA2 and HNF4A [Ref.1], but lowly expressed the large cholangiocyte marker genes of SCTR, CASP9 and I12RB [Refs. 2, 3]. We have added these new data and re-organized their presentation in the revised manuscript.

References:

1. Glaser, S. et al. Differential transcriptional characteristics of small and large biliary epithelial cells derived from small and large bile ducts. *Am. J. Physiol. Gastrointest. Liver Physiol.* **299**, G769-77 (2010).
2. Boyer, J.L. & Soroka, C.J. Bile formation and secretion: An update. *J. Hepatol.* **75**:190-201 (2021).
3. Ueno, Y. et al. Evaluation of differential gene expression by microarray analysis in small and large cholangiocytes isolated from normal mice. *Liver Int.* **23**, 449-459 (2003).
8. *Figure 3I are impossible to analyses. They are just too small and their resolution not good enough. It would be incredibly useful to show a bile duct with containing both KRT19/DUOX2⁺/ACE2⁺ and KRT19/DUOX2⁻/ACE2⁻ cells in control. Indeed, all the*

KRT19+ cholangiocytes seems to express DUOX2 and ACE2.

Response: We are sorry for the confusion and appreciate your advice. Now, we have carefully examined these imaging data. We found that some panels of multiplex IF images were overexposed, resulting in the consequences that all cholangiocytes appeared to be stained with anti-DUOX2 and anti-ACE2 in the original Figure 3I. To address this concern, we have performed additional multiplex IF in these liver sections and adjusted its exposure intensity (new Fig. 3g). Moreover, we also performed RNAScope in liver sections to confirm our observations (new Fig. 3b and Supplementary Fig. 11). As suggested, we now show both CK19⁺DUOX2⁺ACE2⁺ cells and CK19⁺DUOX2⁻ACE2⁻ cells in a bile duct of a control patient liver section (Supplementary Fig. 12). Accordingly, we have added these new data and made corresponding changes in the revised manuscript and supplementary documents.

9. *The data on pIgR are really interesting but lack functional validation. It would be interesting to define if patient immune cells react specifically against cholangiocytes expressing this antigen.*

Response: We appreciate his/her insightful suggestion. We have performed DUOX2⁺ACE2⁺ cholangiocyte-immune cell interaction analysis by using the CellPhoneDB based on receptor–ligand interactions between two cell types/subtypes [Ref.1]. We detected substantial interactions between ligands and receptors expressed

by immune cells and DUOX2⁺ACE2⁺ small cholangiocytes that highly express pIgR antigen (Fig. 5a-c). We have added these new data in the revised manuscript.

References:

1. Efremova, M. et al. CellPhoneDB: inferring cell-cell communication from combined expression of multi-subunit ligand-receptor complexes. *Nat. Protoc.* **15**, 1484-1506 (2020).

Some specific points:

1. *Figure 1H - It should be denoted which comparisons are significant and the full p value provided in the legend*

Response: We are sorry for the confusion. We have made them clear in the revised Fig.1 and the legend.

2. *Text – line 70 - In text they refer to many supplementary tables - this a huge amount of info much not discussed at this stage. The tables should only be referenced when appropriate to aid the reader in following flow of the paper*

Response: We appreciate your advice. Now, we have re-organized these supplementary tables and cited them in the text of the revised manuscript

3. Text – *“Interestingly, there was a unique cholangiocyte cluster (3) in control livers, but a few were detectable in PBC livers (Fig.2C-D).” This need to be reworded - be more specific and clear.*

Response: We have modified the sentence in the revised manuscript. Thanks.

4. Figure 2G - FACS is confusing way it is presented. Lower panel is all Ck19+ as it is a daughter gate of the upper panel. Should not be shown as a quadrant, a single line should partition the ACE+ and ACE2-.

Response: We appreciate his/her advice. Now, we have made such correcteion in the new Fig. 2g in the revised manuscript.

5. Figure 3E and F - are missing scale bars. More images should be shown with examples of small, medium and large ducts shown if possible, so the differences can be appreciated. Scale bars are necessary for this interpretation.

Response: Corrections are made in the revised Figure 3E and F (now as Fig. 3c, d).

Imaging quality is low and appears out of focus, clearer images should be taken.

Response: We have improved the quality of these images in the revised Fig. 3.

6. *Figure 3I - Also missing scale bars and images could be improved*

Response: We appreciate your suggestions. We have added scale bars and improved imaging quality in Figure 3I (now as Fig. 3g).

7. *Figure 4B - Single fluorescent channels should be shown to help with interpretation*

Response: We have made the change in the revised manuscript (Supplementary Fig. 13).

8. *Figure 5C - spatial transcriptomics - wouldn't they expect that the proportion of cell samples on the sections which are hepatocytes to be higher - at least in the control?*

Response: Thank you for your critical comments. Firstly, hepatocytes are extremely sensitive to the processes of experiments, such as dissociation and OCT-embedded method, resulting in hepatocyte death. Secondly, the expression profile of different types of cells obtained in the scRNA-seq was used as a reference to obtain the cell composition of the spatial transcriptome by the method of deconvolution [Ref.1]. It might also decrease the proportion of hepatocytes in ST dots in this process. In addition, hepatocytes consist of ~60% of liver cells [Ref.2]. However, due to their larger size, they make up ~80% of the mass/volume of liver tissue [Ref.3]. Because each hepatocyte takes a larger area, their proportion of the total cell number is lower in the

spatial transcriptome. Of note, spatial transcriptome in our study were mainly used to investigate the interactions between immune cells and cholangiocytes. Moreover, the distribution of hepatocytes in HE staining is also similar to its distribution in spatial transcriptome (Fig. 4c, d). Nevertheless, to avoid the potential misinterpretation of hepatocytes, we have removed all analyzed data related to the proportion of hepatocytes in both scRNA-seq and spatial transcriptome in the revised manuscript.

References:

1. Mantri, M. et al. Spatiotemporal single-cell RNA sequencing of developing chicken hearts identifies interplay between cellular differentiation and morphogenesis. *Nat. Commun.* **12**, 1771 (2021).
2. Kumar, S., Duan, Q., Wu, R., Harris, E.N., Su, Q. Pathophysiological communication between hepatocytes and non-parenchymal cells in liver injury from NAFLD to liver fibrosis. *Adv. Drug Deliv. Rev.* **176**, 113869 (2021).
3. Blouin, A., Bolender, R.P., Weibel, E.R. Distribution of organelles and membranes between hepatocytes and nonhepatocytes in the rat liver parenchyma. A stereological study. *J Cell Biol.* **72**, 441-55 (1977).
9. *Figure 5F - no scale bars. Single channels should be shown at least for the triple stain to help with interpretation.*

Response: We have modified the Figure 5F (now as Fig. 4f) and provided the images of single channels as the Supplementary Fig. 13 of the revision.

10. *Figure 5G - it is unclear to me what this analysis is showing and what N represents.*

This should be explained In rebuttal and in the methods.

Response: We apologize for the confusing interpretation. The Figure 5G (now as Fig. 4g) displayed the distribution of BCR clonotypes in B cell subtypes. We combined the single-cell RNA sequencing and the single-cell BCR sequencing results to count this distribution by the sample. The clonotypes were classified by their frequency of occurrence. N=1: clonotypes existed only in one cell; N=2: clonotypes existed in two cells; N>=3: clonotypes existed in more than 3 cells. The ordinate represented the percentage of these 3 clonotypes categories described above in total clonotypes number of different B cell subtypes. We have added this detailed explanation in the Method section and the corresponding figure legend in the revised manuscript.

Responses to the Reviewer 3:

The manuscript "NCOMMS-21-21721-T" entitled "Unique DUOX2+ACE2+ small cholangiocytes are pathogenic targets of primary biliary cholangitis" presents a study of the chronic autoimmune liver disease called primary biliary cholangitis (PBC). Among the multiple analytical techniques utilized, the authors performed proteomics analyses by data-independent acquisition (DIA) mass spectrometry (MS). Overall, the proteomics experiments and results are well described. The following points are

recommended to be addressed to improve the manuscript:

Response: We greatly appreciate his/her positive comments.

1. *Proteomics is not explicitly mentioned in the abstract.*

Response: Thank you for pointing this out. Now, we have added proteomics in the Abstract section in the revised manuscript.

2. *Please clarify how the individual samples were prepared for the DIA-MS analysis, since the described digestion and fractionation is referring to the pooled samples used to build the DDA library.*

Response: We greatly appreciate his/her professional advice. The procedures for preparing the individual samples for the DIA-MS analysis were described as follows: Firstly, the individual serum samples were solved in SDT buffer (4% SDS, 100 mM DTT, 150 mM Tris-HCl pH 8.0), boiled for 15 minutes and centrifuged at 14000 g for 20 minutes. Secondly, the supernatants were aliquoted and stored at -80 °C. Subsequently, the soluble proteins in the individual serum samples were digested according to the FASP protocol. We have added these details in the Method section of the revised manuscript.

3. *The nanoLC conditions used for the DIA-MS analysis should be included.*

Response: Thank you for the suggestion. The nanoLC conditions for the DIA-MS analysis were described as follows: The peptides were loaded onto a reverse phase trap column (Thermo Scientific Acclaim PepMap100, 100 μm^2 cm, nanoViper C18) connected to the C18-reversed phase analytical column (Thermo Scientific Easy Column, 10 cm long, 75 μm inner diameter, 3 μm resin) in buffer A (0.1% Formic acid) and separated with a linear gradient of buffer B (84% acetonitrile and 0.1% Formic acid) at a flow rate of 300 nl/min controlled by IntelliFlow technology. We have added this description in the Method section in the revised manuscript.

4. The instrument used for DIA-MS analyses can be inferred from the description but it should be clearly specified including the vendor, in the same way the instrument used to build the DDA-MS library was described.

Response: We appreciate his/her advice. Now, we have added the instrument information and described the procedures in details for the DIA-MS analyses and the DDA-MS library generation in the Method section of the revised manuscript.

Responses to the Reviewer 4:

This manuscript identifies a sub-population of human cholangiocytes that specifically express ACE2, DUOX2 and PIGR and is enriched in bile secretion processes. The

authors claim that these cholangiocyte subtypes are the target of autoimmune attack in PBC, giving rise to the associated portal inflammation. They further claim that the mechanism involves the death of these cells through excessive transfer of Iga, the consequent release of PIGR proteins and the stimulation of the production of auto-antibodies against PIGR. They perform proteomic measurements in blood to validate an increase in PIGR and in auto-antibodies against PIGR, supporting the model.

The manuscript is very interesting and proposes a novel mechanism that may underlie PBC. However, some of the key claims in the paper are not sufficiently supported by the data and should either be substantiated with additional experiments and analyses or removed. The following points must be addressed.

Response: We thank his/her positive comments and constructive suggestions.

1. *The authors claim that the ACE2+DUOX2+ cholangiocytes constitute 55% of cholangiocytes in control human livers. This finding is not supported by the data. From our analysis of the single cell 3'-UTR single cell data the cells only appear in one of the two control patients, with very low proportions in the second control patient. This is also consistent with previous human liver single cell atlases, in which these cells appear in extremely low proportions, as can be readily verified in the following browsers:*

<http://human-liver-cell-atlas.ie-freiburg.mpg.de/>,

<https://shiny.igmm.ed.ac.uk/livercellatlas/>,

<https://itzkovitzwebapps.weizmann.ac.il/webapps/home/session.html?app=HumanLiv>

erBrowser. The authors should present a table with the proportions of the ACE2+DUOX2+ cells in each patient as well as UMAPS colored by patient. They should also at least hypothesize why the abundances of these cells is so variable and why have they not been observed in previous human liver atlas studies (different cohorts? Different cell extraction protocols)?

Response: We greatly appreciate his/her critical comments and insightful advice. To address these concerns, we have performed additional FACS experiments and re-analyzed the 5'-scRNAseq data. Firstly, our new FACS data by using fluorescein conjugated isotype control antibodies indicated that DUOX2⁺ACE2⁺ small cholangiocytes accounted for 18.7% or 22.2% of cholangiocytes in human control livers or normal mouse livers, respectively (Fig. 2g), similar to its proportion (18.63% of cholangiocytes) analyzed by the 5'-scRNAseq in human control livers (Fig. 2d and Supplementary Table 6). As suggested, we have provided the Supplementary Tables 6 and 7 to display the proportion of DUOX2⁺ACE2⁺ small cholangiocytes in each patient or each group. Regarding the abundance variation of these cholangiocytes, it is likely to be attributed to human control livers resected from those with different types of liver diseases, including hepatic hemangioma and intrahepatic duct stone although they had normal levels of liver function tests. Moreover, in our study, we found that only PBC preferably targeted DUOX2⁺ACE2⁺ small cholangiocytes in the bile ducts (Fig. 3g). This characteristic might contribute to the finding that the proportion of DUOX2⁺ACE2⁺ small cholangiocytes was not altered in other liver diseases, such as

obstructive cholestasis, NASH, etc. (Fig. 3g) and why DUOX2⁺ACE2⁺ small cholangiocytes were not found in previous human liver atlas. Accordingly, we have added these new data in the revised manuscript and supplementary documents.

2. A related note – the in-situ validations of these cells is not convincing. For example, Figure 3E seem to show 100% of cholangiocyte positive for the ACE2 and DUOX2 proteins. There are no scale bars in any images, the images are low resolution. Please include high magnification high resolution blow-ups. It is unclear if the positive cells are interspersed between the negative cells or rather cluster in distinct bile ducts, please clarify.

Response: To address these concern, we have carefully examined our experimental data and performed additional experiments. We recognized that some panels of multiplex IF images were overexposed, leading to this phenomenon that all cholangiocytes appeared to be stained with anti-DUOX2 and anti-ACE2 (the original Figure 3E). To improve the quality of multiplex IF images, we have repeated the multiplex IF in these liver sections and adjusted their exposure intensity in the revised Fig. 3c, d. Moreover, we also performed RNAScope in liver sections to confirm our observations (the new Fig. 3b, Supplementary Figs. 11 and 12). Interestingly, we found that DUOX2⁺ACE2⁺ cholangiocytes were interspersed with negative cholangiocytes in the bile ducts by both multiplex IF and RNAScope analyses (Fig. 3b, c, Supplementary Fig.12). We have presented these new data and made corresponding changes in the

revised manuscript.

Most importantly, little information is given on the antibody validation. The authors should perform RNA in-situ hybridizations for the relevant genes of interest, or conversely use well-validated antibodies. ACE2 and DUOX2 are expressed in intestinal villi, and so showing staining on such a tissue with established expression of the proteins is important for such validation.

Response: We have performed RNAScope (the new Fig. 3b, Supplementary Figs.11 and 12) to confirm our observations using the multiplex IF analysis. Furthermore, we have also validated the specificity of the ACE2 or DUOX2 antibodies in human or mouse intestinal villi (Supplementary Fig.17), according to your helpful advice. We have added these new data in the revised manuscript and supplementary documents.

Figure 5F – again, in-situ quantification is problematic, no scale bars, not clear if CK19 has a high background or if we are seeing multiple bile ducts. Single molecule RNA in-situ hybridization (e.g. RNAScope) has substantially lower background and is therefore the gold standard.

Response: We appreciate his/her helpful advice. As described above, we have performed RNAScope in human liver sections (the new Fig. 3b, Supplementary Figs. 11 and 12). Moreover, to improve the quality of staining images of DUOX2⁺ACE2⁺ cholangiocytes, I have repeated the multiplex IF labelling of CK19, DUOX2 and ACE2

in the serial frozen liver sections that were previously used for spatial transcriptomics (new Fig. 4f). In addition, the specificity of the CK19 antibody was also validated in human liver sections by using a negative IgG control (Supplementary Fig.17). We have added these new data and modified the related figures in the revised manuscript.

3. *Mouse experiments – the authors claim that ablation of Ace2+Duox2+ cholangiocytes promotes cholestasis. For this direction to be relevant to the establishment of the functional importance of these cells, the authors should first demonstrate the existence of these cells in mouse liver via one of many available single cell mouse liver atlases (e.g. <https://www.sciencedirect.com/science/article/pii/S1097276519305830#app2>). From brief analysis I see a very small minority of mouse cholangiocytes expressing these markers. In addition, the in-situ validation of the existence of the Duox2+ mouse cholangiocytes must be expanded (Figure 4B shows one example and quantification is unclear). A fundamental problem with the ablation experiment is that Duox2 and Ace2 are highly expressed in the intestinal epithelium. If the ablation yields decrease in intestinal epithelial cells, this would lead to systemic inflammation. If this piece of evidence is to be used, the authors must demonstrate that intestinal epithelial cells are not damaged and that the Duox2+ cholangiocytes are lost upon ablation. In my view, this direction is problematic and might need to be removed from the paper (the other findings of the existence of the human ACE2+DUOX2+ in even a small sub-population and its potential role in human PBC pathology is interesting enough in my view,*

considering that the other points are addressed).

Response: We fully agree with the reviewer' critical comments and appreciate your suggestions. Accordingly, we have removed all data related to mouse models in the revised manuscript.

4. Spatial transcriptomics – Analysis is problematic, hepatocytes seem to take up around 20% of the spot transcriptomes, this cannot be, it should approach 80%, please clarify or analyze differently.

Response: Thank you for your critical comment. In general, hepatocytes consist of ~60% of liver cells [Ref.1]. However, due to their larger size, they make up ~80% of the mass/volume of liver tissue [Ref.2]. Because each hepatocyte takes a larger area, their proportion of the total cell number is lower in the spatial transcriptome. Moreover, hepatocytes are extremely sensitive to the processes of experiments, such as dissociation and OCT-embedded method, resulting in hepatocyte death. In addition, the expression profile of different types of cells obtained in the scRNA-seq was used as a reference to obtain the cell composition of the spatial transcriptome by the method of deconvolution [Ref.3]. It might also decrease the proportion of hepatocytes in ST dots in this process. To further make sure this is the case, we have re-analyzed our spatial transcriptomics data and 5'-scRNA-seq data, the results indicate that the proportion of hepatocytes in liver cells remains relatively low as previously described in sc-RNAseq data [Refs.4, 5]. In addition, we have carefully reviewed the literature for

the spatial transcriptome in the liver. However, there is no report that describes the proportion of hepatocytes in ST dots [Refs.6-8]. Moreover, the distribution of hepatocytes in HE staining was similar to its distribution in spatial transcriptome (Fig. 4c, d). Of note, spatial transcriptome in our study was mainly used to investigate the interactions between immune cells and cholangiocytes. Therefore, to reduce any potential misinterpretation, we have removed all analyzed data related to the proportion of hepatocytes in both scRNA-seq and spatial transcriptome analysis. We have made corresponding changes in the revised manuscript and supplementary documents.

References:

1. Kumar, S. et al. Pathophysiological communication between hepatocytes and non-parenchymal cells in liver injury from NAFLD to liver fibrosis. *Adv. Drug Deliv. Rev.* **176**, 113869 (2021).
2. Blouin, A., Bolender, R.P. & Weibel, E.R. Distribution of organelles and membranes between hepatocytes and nonhepatocytes in the rat liver parenchyma. A stereological study. *J Cell Biol.* **72**, 441-455 (1977).
3. Mantri, M. et al. Spatiotemporal single-cell RNA sequencing of developing chicken hearts identifies interplay between cellular differentiation and morphogenesis. *Nat. Commun.* **12**, 1771 (2021).
4. Sun, X.J. et al. Transcriptional switch of hepatocytes initiates macrophage recruitment and T cell suppression in endotoxemia. *J. Hepatol.* **S0168-8278**, 00136-

- 2 (2022).
5. Aizarani, N. et al. A human liver cell atlas reveals heterogeneity and epithelial progenitors. *Nature* **572**, 199-204 (2019).
 6. Guilliams, M. et al. Spatial proteogenomics reveals distinct and evolutionarily conserved hepatic macrophage niches. *Cell* **185**, 379-396 (2022).
 7. Hildebrandt, F. et al. Spatial Transcriptomics to define transcriptional patterns of zonation and structural components in the mouse liver. *Nat. Commun.* **12**, 7046 (2021).
 8. Gao, S. et al. Identification of HSC/MPP expansion units in fetal liver by single-cell spatiotemporal transcriptomics. *Cell Res.* **32**, 38-53 (2022).

5. *Figure 5G analysis is unclear, please elaborate.*

Response: Thank you for pointing this out. As suggested, we have added the detailed explanation in the Method section and the figure legend in the revised manuscript.

Reviewers' Comments:

Reviewer #1:

Remarks to the Author:

The authors made comments on the reviewers points

Reviewer #3:

Remarks to the Author:

The authors have addressed the concerns regarding the DIA methods.

Reviewer #4:

Remarks to the Author:

The authors have done a great job in addressing all of my comments.

Nature Communications NCOMMS-21-21721B

Title: Unique DUOX2⁺ACE2⁺ small cholangiocytes are pathogenic targets for primary biliary cholangitis

Point-by-Point Response to the Reviewers

Response to the Reviewer #1:

The authors made comments on the reviewers points.

Response: We greatly appreciate the reviewer for his/her positive and insightful comments which helped us significantly improve our manuscript.

Response to the Reviewer #3:

The authors have addressed the concerns regarding the DIA methods.

Response: Thanks very much for the reviewer's professional comments which allowed us to improve the manuscript significantly.

Response to the Reviewer #4:

The authors have done a great job in addressing all of my comments.

Response: We greatly appreciate the reviewer for his/her positive and constructive comments which improved our manuscript significantly.